# Scrutinizing the Antimicrobial and Antioxidant Potency of European Cranberry Bush (*Viburnum opulus* L.) Extracts

**Karina Juhnevica-Radenkova** [1], **Inta Krasnova** [1], **Dalija Seglina** [1], **Sandra Muizniece-Brasava** [2], **Anda Valdovska** [3] and **Vitalijs Radenkovs** [1,3,*]

1   Institute of Horticulture (LatHort), LV-3701 Dobele, Latvia; karina.juhnevica-radenkova@lbtu.lv (K.J.-R.); inta.krasnova@lbtu.lv (I.K.); dalija.seglina@lbtu.lv (D.S.)
2   Food Institute, Faculty of Agriculture and Food Technology, Latvia University of Life Sciences and Technologies, LV-3004 Jelgava, Latvia; sandra.muizniece@lbtu.lv
3   Research Laboratory of Biotechnology, Latvia University of Life Sciences and Technologies, LV-3004 Jelgava, Latvia; anda.valdovska@lbtu.lv
*   Correspondence: vitalijs.radenkovs@lbtu.lv

**Abstract:** In the process of considering the documented health benefits of *Viburnum opulus* L. (*V. opulus*), including its anti-inflammatory and antioxidant activities, the present study was designed to qualitatively and quantitatively evaluate the biochemical profile and antimicrobial potency of four commercially available *V. opulus* extracts. These extracts were obtained from its flowers, bark, berries, and a mixture thereof by cold ultrasound-assisted extraction. An examination of the *V. opulus* extracts indicated a relative abundance of group compounds, such as phenolics, flavonoids, tannins, and anthocyanins, which are responsible for antioxidant activity (AOA). The widest range in all of the four group compounds was detected in the *V. opulus* extract sourced from berries, whereas the narrowest range was found in those obtained from flowers. The HPLC-ESI-TQ-MS/MS technique displayed relative fluctuations in the concentrations of individual amino acids (AAs) over the four *V. opulus* extracts. The prevalence of proline was marked in the flower-derived extract, which made up 63.3% of the total AAs, while aspartic and glutamic acids dominated in the berry-derived extract by contributing up to 29.2 and 24.4% to the total AA content, respectively. Profiling of the individual phenolic compounds disclosed the superiority of chlorogenic acid (up to 90.3%) in the berry and mixed extracts, as well as catechin (up to 57.7%) and neochlorogenic acid (11.1%) in the bark extract, which conveyed a remarkable contribution toward antimicrobial activity. The lowest content of individual phenolics was found in the flower extract. Owing to its substantially denser bioactive composition, the *V. opulus* berries and bark extracts exhibited markedly better AOA, which was pinpointed by three independent methods, i.e., DPPH•, FRAP, and ABTS•+, than those obtained from flowers or a mixture of *V. opulus* morphological parts. As part of the antimicrobial activity testing, the *V. opulus* extracts exhibited outstanding inhibitory activity and a homeopathic mode of action. The *V. opulus* extracts obtained from a mixture, bark, and berries were more active against 8 out of 19 selected test microorganisms at minimum inhibitory concentration (MIC) values that ranged from 0.24 to 0.49 $\mu L\ mL^{-1}$. Overall, the extracts of *V. opulus* were found to be effective against Gram-positive and Gram-negative bacteria. However, their conceivable exploitation as functional or pharmaceutical ingredients must be further clarified within in vivo models.

**Keywords:** antibacterial activity; catechins; chlorogenic acid; cramp bark bioactives; minimum inhibitory concentration; resistance

## 1. Introduction

In the Middle Ages, men presented bouquets of European cranberry bush (*Viburnum opulus* L.) (*V. opulus*) flowers to create positive feelings for women and manifest attention [1]. Today, *V. opulus* (family *Caprifoliaceae*) is known as the snowball fruit, European cranberry bush, guelder rose, guilder rose, water elder, or cramp bark; it is widely

cultivated around the world for ornamental or pharmaceutical purposes and less as edible fruit. The genus *Viburnum* consists of over 200 species, among which the berries of *V. opulus* have a bitter and astringent taste. The compounds responsible for bitterness and astringency make them unpleasant in terms of flavor and negatively affect the sensory quality of the final product [2–4]. The use of alcohol tinctures made from the bark of the *V. opulus* root, shrub, and its limbs for medicinal purposes dates back to 1882 by Purdy [5], where it was used as a remedy to treat dysmenorrhea and uterine pain. According to the available data, *V. opulus* fruits have been used by native inhabitants in northern parts of the United States, Canada, and Europe as uterine relaxant remedies and for eye disorders [6]. More recent scientific findings imply that *V. opulus* has medicinal properties for treating colds, coughs, tuberculosis, rheumatic aches, and ulcers, as well as stomach and kidney problems [7,8]. With the development of advanced technologies and analytical tools, a great deal of attention has been paid to the detailed analysis of *V. opulus* morphological parts and to elucidating their components' ability to treat various ailments within epidemiological and clinical trials with different therapeutic approaches. In some of these studies, affirmative effects on colon cancer [9], testis and sperm damage [10], inflammations [11], endometriosis [12], gastrointestinal mucosal damage [13], and kidney stone formation [14] are being reflected. Additionally, it has been well documented that the fruits of *V. opulus* serve as ubiquitous reactive oxygen (ROS) and reactive nitrogen species (RNS) scavengers, and their apparent antioxidant abilities have also been well elucidated [2,13,15]. Altun et al. [16] emphasized the implication of the *V. opulus* morphological parts, indicating the outstanding antioxidant activity (AOA) of branch and leaf extracts from it when prepared by maceration in cold distilled water. Furthermore, Dursun et al. [17] underlined the significant impact of extraction methods—such as ultrasound-assisted, microwave-assisted, Soxhlet, and solvent-based—on the yield and quality of *V. opulus* extracts. A recent study by Polka et al. [18], who conducted a histochemical analysis of *V. opulus*, revealed that the ethanol extracts of such morphological parts as bark, flowers, and fruit profusely contain various constituents, including flavan-3-ols (epicatechin and procyanidin B1), flavonols (quercetin), and hydroxycinnamates (chlorogenic and cryptochlorogenic acids). Among 13 of the phenolic compounds detected, the authors emphasized the relative abundance of flavan-3-ols in the 70% bark-derived ethanolic extracts that, among others, contributed to AOA the most. Contrary results have been reported by Karaçelik et al. [19], who analyzed the antioxidant components of the *V. opulus* juice, methanol, and acetonitrile extracts of seeds and skin. The exceptional composition of *V. opulus* juice was underlined, showing the presence of 19 phenolic compounds, among which coumaroylquinic acid, chlorogenic acid, procyanidin B2 (dimer), and procyanidin trimer were dominant. Earlier, Velioglu et al. [3] also revealed the presence of 13 phenolic compounds in the juice of *V. opulus*, where chlorogenic acid was established as a prevalent compound that accounted for 54% or 2037 mg kg$^{-1}$ of the total phenolics. Kraujalyte et al. [20] made a similar observation, indicating chlorogenic acid's superiority over other phenolics (from 540 to 6939 mg kg$^{-1}$) among five of the *V. opulus* genotypes tested. The most recent studies on antimicrobial activity have indicated the exceptional antimicrobial activity of *V. opulus* juice [21–23]. The relevance of this statement is becoming increasingly apparent as the problem associated with the spread of microorganisms that are resistant to topical antibiotics and therapies is turning from a local concern to a global one [24].

The overuse of antibiotics is a primary cause leading to the development of antimicrobial resistance [25]. As the issue of unwarranted and excessive use of antibiotics and the proliferation of microorganism resistance is exacerbated, scientists and medical microbiologists are sounding the alarm for a solution to be addressed. For preventive purposes, it is recommended to use herbal-derived formulations as an exceptional alternative to reduce the risk of morbidity from viral diseases. The recent findings in research into the antimicrobial properties of *V. opulus* fruits indicate their effectiveness in combating pathogens. The abundance of chlorogenic acid and procyanidins in *V. opulus* affects the rigidity and permeability of bacterial cell walls and membranes, disrupting their integrity and reducing

ion exchange and ATF synthesis [26]. This observation is reinforced by the study of Wu et al. [27], who highlighted the interaction of chlorogenic acid with the membrane lipid and protein layers of *Staphylococcus aureus*, thereby causing severe morphological changes that entailed the leakage of intracellular constituents and eventual cell apoptosis. A positive effect of chlorogenic acid was demonstrated by Ji et al. [28], who highlighted the upregulation of interleukin-10 (IL-10) and interleukin-10 receptors (IL-10RA) in bovine mammary epithelial cells as a response to *S. aureus*-induced inflammation. The effectiveness of *V. opulus* fruit extract in inhibiting the growth of *Aeromonas hydrophila* ATCC 7965, *Bacillus cereus* FMC 19, *Enterobacter aerogenes* CCM 2531, *Escherichia coli* DM, *Klebsiella pneumoniae* FMC 5, *Proteus vulgaris* FMC 1, *Pseudomonas aeruginosa* ATCC 27853, *Salmonella typhimurium*, *S. aureus* Cowan 1, and *Yersinia enterocolitica* EU has been proven [29]. The antimicrobial activity of *V. opulus* juice was also emphasized during analyses of the antimicrobial potential against *Salmonella* Agona, *B. subtilis*, *Listeria monocytogenes*, *Enterococcus faecalis*, *Micrococcus luteus*, and *S. epidermidis* [21].

Currently, limited information is available on the effective concentrations of specific *V. opulus* extracts that can inhibit or delay the growth of pathogens. Most studies have focused on antimicrobial activity probing, which has been conducted using the Kirby–Bauer disk diffusion susceptibility method. Thus, this study's primary goal was to examine four commercially available *V. opulus* extracts that were obtained from (1) flowers, (2) a combination of morphological parts (i.e., flowers, berries without seeds, leaves, buds, and bark), (3) bark, and (4) berries without seeds. To achieve this, a qualitative and quantitative analysis of the bioactive profiles and antimicrobial susceptibility test (MIC analysis) of reference test cultures were conducted, and the lowest concentrations that could inhibit the growth of microorganisms were determined.

## 2. Materials and Methods

### 2.1. Chemicals, Standards, and Reagents

Commercial standards of amino acid (AA) mixture (Ref. AAS18); phenolic compounds, i.e., caffeic acid; *trans*-isomer of ferulic acid, vanillic acid, vanillin, *p*-coumaric acid, (−)-epicatechin, (±)-catechin, gallic acid, sinapic acid, syringic acid, protocatechuic acid, neochlorogenic, and chlorogenic acids; rutin; kaempferol; isorhamnetin; luteolin-7-*O*-glucoside; rhamnetin; quercetin; glucose; fructose; sucrose; ribose; 96% ethanol (EtOH); dimethyl sulfoxide (DMSO) (purity 99.7%); and resazurin sodium salt were purchased from Sigma-Aldrich Chemie Ltd. (Steinheim, Germany). Methanol (MeOH), acetonitrile (MeCN), and formic acid (HCOOH) (puriss p.a., ≥99.9%) of a liquid chromatography-mass spectrometry (LC-MS) grade were obtained from Merck KGaA (Darmstadt, Germany). Ultrapure water (UPW) was produced using the reverse osmosis "PureLab Flex Elga" water purification system (Veolia Water Technologies, Paris, France).

### 2.2. Sample Information

Four commercially available *Viburnum opulus* L. extracts (BestBerry Ltd., Auce, Latvia) obtained from biologically grown fruit and their morphological parts, i.e., flowers (1 Flowers); a mixture of morphological parts (i.e., flowers, berries without seeds, leaves, buds, and bark) (2 Mix); bark (3 Bark); and berries without seeds (4 Berry); were purchased on February 2024 from the local pharmacy. The labels on the extracts indicated that they would expire on January 2025. The details of the extraction process are not disclosed in this study as they refer to technical confidential information and trade secrets. The raw materials used for extract production were grown in Latvia, specifically in the city of Auce (56°27′20.1″ N 22°54′23.0″ E). All raw materials were stored by the manufacturer of the extract using vacuum bags that were impermeable to air and moisture in a frozen state at −18 ± 1 °C until further processing and use for a maximum of 12 months. Ultrasonic treatment and cold water as a solvent were used to prepare the extracts.

### 2.3. Preparation of the Extracts for an Analysis of the Group Compounds and Antioxidant Activity

In this study, 1 mL of *V. opulus* extract was introduced into a 50 mL conical centrifuge tube (Sarstedt AG & Co., KG, Nümbrecht, Germany), and 25 mL of acidified 80% MeOH (MeOH:H$_2$O:HCOOH ratio 80:19:1 *v/v/v*) was then added. The obtained mixture was intensively Vortex mixed for 2 min with a "Vortex REAX top" (Heidolph, Schwabach, Germany), which was followed by ultrasonic treatment at 50 kHz with an output wattage of 360 W for 30 min at 25 ± 1 °C using an "Ultrasons" ultrasonic bath (J.P. Selecta®, Barcelona, Spain). Afterward, the mixture was centrifuged at 3200× *g* in an "Eppendorf 5804 R" centrifuge (Eppendorf AG, Hamburg, Germany) for 10 min at 20 °C, and the top organic layer was filtered through a "Whatman® Grade 6" filter (Cytiva, Marlborough, MA, USA). The clear filtrate was then used for spectrophotometric studies to determine the phenolic (TPC), flavonoid (TFC), and tannin (TTC) contents, as well as the antioxidant activity, by DPPH• and ABTS•+ methods and by ferric-reducing antioxidant power (FRAP).

### 2.4. Preparation of the Extracts for an Analysis of the Anthocyanins

The total anthocyanins (TAC) were extracted using the methodology provided by Krasnova et al. [30]. Briefly, 3 mL of *V. opulus* berry (4) or 5 mL of flowers (1), mixed (2), or bark (3) extract was transferred into a 50 mL centrifuge tube, and 30 mL of acidified 96% ethanol (EtOH:1N HCl ratio 85:15 *v/v*) was then introduced. The obtained mixtures were, respectively, thoroughly mixed and subjected to ultrasonic treatment for 30 min at room temperature 22 ± 1 °C. The obtained extracts were then centrifuged at 3200× *g* for 10 min at 20 °C, and the top organic layers were then filtered through a "Whatman® Grade 6" filter.

### 2.5. Spectrophotometric Studies

#### 2.5.1. Determination of the Phenolic Content

The total phenolic content (TPC) was determined using the colorimetric Folin–Ciocalteu method in accordance with Singleton et al. [31]. Briefly, a 0.5 mL aliquot of each *V. opulus* extract, or the standard gallic acid (GA), was mixed with 2.5 mL of 10-fold diluted Folin–Ciocalteu reagent, which was then followed by the addition of 2 mL 7.5% Na$_2$CO$_3$ with a subsequent incubation for 30 min at room temperature (22 ± 1 °C). Finally, the absorbance was measured at a wavelength of 760 nm using a Shimadzu series visible spectrophotometer "UV-1800" (Shimadzu Corp., Kyoto, Japan). The results were expressed as a mg gallic acid equivalent per 100 mL$^{-1}$ on a fresh weight basis (mg GAE 100 mL$^{-1}$ FW). Least squares regression analysis was used to determine the absorbance intensities against increasing standard concentrations to obtain calibration linearity (y = ax ± b, where "b" is the slope and "a" is the intercept point at the "y" axis). The method's precision was estimated by a triplicate analysis of standard solutions at seven calibration levels (Supplementary Table S1).

#### 2.5.2. Determination of the Flavonoid Content

The total flavonoid content (TFC) was determined using the methodology described by Yang et al. [32] with slight modifications. Briefly, 1 mL of *V. opulus* extract was transferred into a 15 mL centrifuge tube, and 2 mL of UPW and 0.15 mL of 5% NaNO$_2$ were added. The mixture was thoroughly mixed and allowed to react for 6 min. Then, 0.15 mL of 10% AlCl$_3$ was added, thoroughly mixed, and allowed to react for an additional 6 min. Finally, 2 mL of 1 N NaOH and 4.7 mL of UPW were added to adjust the total volume to 10 mL. Then, the absorbance was measured after 15 min at a wavelength of 510 nm. The results were expressed as a mg catechin equivalent per 100 mL$^{-1}$ on a fresh weight basis (mg CE 100 mL$^{-1}$ FW) (Supplementary Table S1).

#### 2.5.3. Determination of the Tannin Content

The total tannin content (TTC) was determined according to the protocol of Ojha et al. [33] with some modifications. Briefly, 0.5 mL of *V. opulus* extract was transferred

into a 15 mL centrifuge tube, and either 3 mL (flowers, mix, or bark extract) or 5 mL (*V. opulus* berry extract) of UPW, as well as 1 mL of 0.1M $FeCl_3$ and 1 mL of 8 mM $K_3Fe(CN)_6$, were added. The obtained solution was adjusted with UPW to the final volume of 10 mL, which was then mixed well and allowed to stand for 10 min at $22 \pm 1\ ^\circ C$ in the dark. The absorbance was measured at 720 nm. The actual tannin concentrations were calculated based on the optical absorbance values obtained for the standard solutions. The results were expressed as the mg catechin (CE) equivalent per 100 $mL^{-1}$ on a fresh weight basis (mg CE 100 $mL^{-1}$ FW) (Supplementary Table S1).

### 2.5.4. Determination of the Anthocyanin Content

The total anthocyanin content (TAC) was determined as was performed by Sadowska et al. [34]. The absorbance of the obtained extracts was measured at a wavelength of 510 and 700 nm. The results were expressed as a mg cyanidin-3-*O*-glucoside equivalent (CGE) per $mL^{-1}$ on a fresh weight basis (mg CGE 100 $mL^{-1}$ FW).

### 2.6. The HPLC-RID Conditions for Carbohydrate Analysis

A quantitative analysis of free mono- and disaccharides in *V. opulus* extracts was conducted using a "Waters Alliance" high-performance liquid chromatography (HPLC) system (Model No. e2695), which was coupled to a 2414 refractive index detector (RID) and a 2998 column heater (Waters Corporation, Milford, MA, USA), following the methodology described by Radenkovs et al. [35].

### 2.7. Preparation of Amino Acids for LC-ESI-TQ-MS/MS Analysis

The isolation and purification of amino acids (AAs) from *V. opulus* extracts were performed following liquid–liquid extraction. Briefly, 1 mL of *V. opulus* extract with an accuracy of $\pm 0.01$ mL was transferred into 15 mL conical centrifuge tubes, and 9 mL of 20% acidified MeCN (MeCN:$H_2O$:HCOOH ratio 20:79:1 *v/v/v*) was then added. Afterward, the mixture was subjected to a 1 min intensive Vortex mixing using the "ZX3" vortex mixer (Velp® Scientifica, Usmate Velate, Italy), which was followed by ultrasonic treatment at 50 kHz with an output power of 360 W for 30 min and a temperature of $20 \pm 1\ ^\circ C$ using an "Ultrasons" ultrasonic bath (J.P. Selecta®, Barcelona, Spain). Afterward, the obtained extracts were centrifuged at $10,280 \times g$ for 12 min at $4 \pm 1\ ^\circ C$ in a "Hermle Z 36 HK" centrifuge (Hermle Labortechnik, GmbH, Wehingen, Germany). The filtration of the collected supernatant was performed using a 0.20 µm hydrophilized polytetrafluoroethylene (H-PTFE) membrane filter (Macherey-Nagel GmbH & Co. KG, Dueren, Germany). The pre-concentration and purification of AAs was conducted by subjecting the obtained filtrate (1 mL) to a reverse spin using an "Amicon® Ultra" 2 mL centrifugal filtration device (Ultracel®, molecular weight cut-off 3 Kilodaltons, Merck Millipore Ltd., Cork, Ireland) (Figure 1).

The centrifugation of loaded samples was performed for 20 min at $19 \pm 1\ ^\circ C$ in $3200 \times g$ conditions. The obtained filtrate fractions were kept at $-18 \pm 1\ ^\circ C$ until required for further analysis and use, or for a maximum of 12 h.

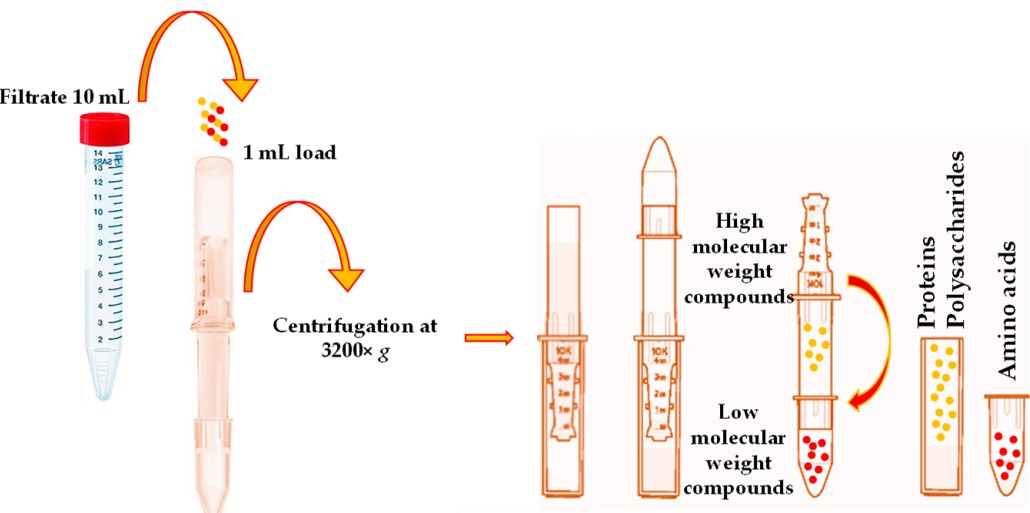

**Figure 1.** Pre-concentration and purification of amino acids by an Amicon® Ultra 2 mL centrifugal filtration device through a 3 Kilodalton cut-off.

### 2.8. The LC-ESI-TQ-MS/MS Analytical Conditions for Amino Acids

The chromatographic analysis of the AAs recovered from commercially available *V. opulus* extracts was performed on a "Shimadzu Nexera UC" series supercritical fluid extraction–supercritical fluid chromatography–liquid chromatography (SFE-SFC-LC) system (Shimadzu Corporation, Tokyo, Japan), which was coupled to a triple quadrupole mass-selective detector (TQ-MS-8050, Shimadzu Corporation, Tokyo, Japan) with an electrospray ionization interface (ESI). A sample of 0.03 μL L$^{-1}$ was injected onto a reversed-phase "Discovery® HS F5-3" column (3 μm, 150 × 2.1 mm, Merck KGaA, Darmstadt, Germany) operating at 40 °C and a flow rate of 0.25 mL min$^{-1}$. The mobile phases used were acidified UPW (H$_2$O:HCOOH ratio 99.9:0.1 *v/v*) (A) and acidified MeCN (MeOH:HCOOH ratio 99.9:0.1 *v/v*) (B). The program for a stepwise gradient elution of the mobile phase B for 20 min was implemented as follows: T$_{0 min}$ = 5%, T$_{5min}$ = 30%, T$_{11min}$ = 60%, T$_{12min}$ = 80%, and T$_{12min}$ = 5%. Finally, re-equilibration for 3 min was conducted after each analysis following the initial gradient conditions. The MeCN injections were included as a blank run after each sample to avoid the carry-over effect. Data were acquired using "LabSolutions Insight" LC-MS software version 3.7 SP3 (workstation), which was also used for instrument control and processing. Ionization was applied in the positive ion polarity mode in this study, while the data were collected in the profile and centroid modes with a data storage threshold of 5000 absorbance for MS. The operating conditions were as follows: a detector voltage of 1.98 kV, a conversion dynode voltage of 10 kV, an interface voltage of 4 kV, an interface temperature of 300 °C, a desolvation line temperature of 250 °C, a heat block temperature of 400 °C, nebulizing gas argon (Ar, purity 99.9%) at a flow of 3 L min$^{-1}$, heating gas carbon dioxide (CO$_2$, purity 99.0%) at a flow of 10 L min$^{-1}$, and drying gas nitrogen (N$_2$, separated from air using a nitrogen generator system from "Peak Scientific Instruments Ltd." (Inchinnan, Scotland, UK), purity 99.0%) at a flow of 10 L min$^{-1}$. All AAs were observed in the programmed and optimized multiple reaction monitoring (MRM) mode. The MRM transitions, collision energy, Q1, Q3, and dwell time for the AAs are detailed in Supplementary Table S2. The representative chromatographic separation of the amino acids in the four *V. opulus* extracts is shown in Supplementary Figures S1–S4.

Preparation of the Standard Stock Solution

A stock solution containing 0.025 uM mL$^{-1}$ AAs was prepared in 10 mL of a 20% acidified MeCN solution (MeCN:H$_2$O:HCOOH ratio 20:79:1 *v/v/v*). The AAs were quantified by injecting 3 μL at 15 °C of a calibration solution with a range of 0.075 to 2.5 μM L$^{-1}$ (Figure 2).

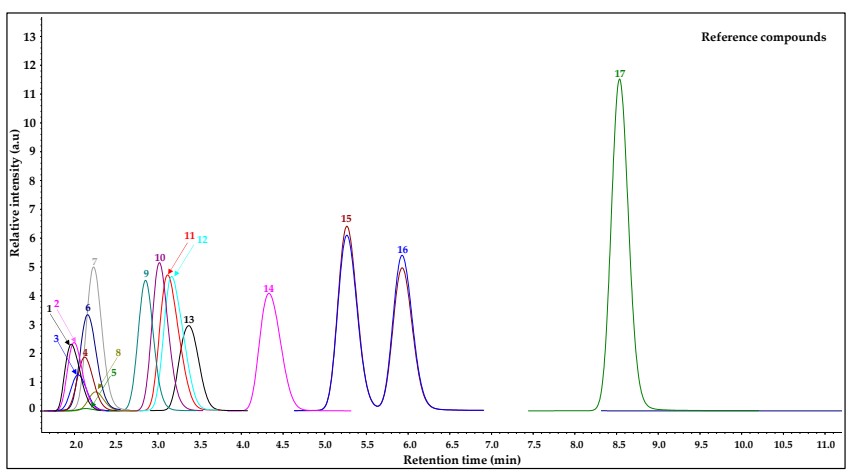

**Figure 2.** The extracted ion chromatogram (EIC) in the multiple reaction monitoring (MRM) represents the profile of 17 multiple amino acids of a standard mixture at the concentration of 2.5 $\mu$M L$^{-1}$ (except cystine 1.25 $\mu$M L$^{-1}$). Note: 1—cystine; 2—aspartic acid; 3—serine; 4—threonine; 5—glycine; 6—glutamic acid; 7—proline; 8—alanine; 9—histidine; 10—lysine; 11—valine; 12—arginine; 13—methionine; 14—tyrosine; 15—isoleucine; 16—leucine; and 17—phenylalanine.

### 2.9. Solid-Phase Extraction of the Free Phenolics for Analysis by LC-ESI-TQ-MS/MS

A solid-phase extraction (SPE) technique, i.e., the isolation and purification of bioactive compounds from the *V. opulus* extracts, was performed following the protocol provided by "Supelco" with minor modifications. Briefly, 1 mL of extract in triplicate was placed in 15 mL conical centrifuge tubes, and 9 mL of acidified 30% MeOH (MeOH:H$_2$O:HCOOH ratio 30:69:1 $v/v/v$) was introduced. Afterward, the obtained mixture was subjected to a 1 min intensive Vortex mixing followed by centrifugation at 10,280$\times$ *g* for 10 min at 20 $\pm$ 1 °C. After centrifugation, the top organic layer was separated and filtered through a 0.20 $\mu$m H-PTFE membrane filter. Purification of the phenolic compounds was approached by SPE "Supel$^{\text{TM}}$-Swift HLB" (57492-U) (Supelco, Bellefonte, PA, USA), which was column-packed with a hydrophilic-modified, styrene-based sorbent (50–70 $\mu$m, 80–200 Å, and 60 mg 3 mL). A steady flow (1.0 $\pm$ 0.2 mL min$^{-1}$) during the analyte desorption was delivered by a "Chromabond$^{\circledR}$ SPE" (Düren, Germany) SPE vacuum manifold with an adjusted pressure of 3.38 $\times$ 10$^{-3}$ Pa. The conditioning/equilibration of the SPE column was performed by a 1 bed volume (3 mL) of pure MeOH, which was followed by a 1 bed volume of acidified UPW (H$_2$O:HCOOH ratio 99:1 $v/v$). The loaded extract (3 mL) was washed with 2 bed volumes of UPW. The flow-through fractions were collected for a qualitative and quantitative chromatographic analysis of the phenolic compound and saccharide presence. A 1 mL acidified MeOH solution (MeOH:HCOOH ratio 99:1 $v/v$) was used to elute the phenolic compounds from the sorbent. The resulting fractions were collected and analyzed with an LC-ESI-TQ-MS/MS system.

### 2.10. The LC-ESI-TQ-MS/MS Analytical Conditions for Individual Phenolic Compounds

The analysis was approached using the same SFE-SFC-MS system mentioned above for AA analysis. Chromatographic separation of the phenolic compounds was carried out using a reversed-phase (RP) "Shim-pack UC-RP" column (5 $\mu$m, 250 $\times$ 4.6 mm; Tokyo, Japan) operating at 45 °C and a flow rate of 1 mL min$^{-1}$. The mobile phases used were an acidified UPW (H$_2$O:HCOOH ratio 99:1 $v/v$) (A) and acidified MeOH (MeOH:HCOOH ratio 99:1 $v/v$) (B). The compounds were separated within 18 min using the stepwise gradient elution program for mobile phase B as follows: $T_{0\text{ min}}$ =5%, $T_{5\text{min}}$ = 10%, $T_{12-15\text{min}}$ = 60%, and $T_{18\text{min}}$ = 10%. Furthermore, MeOH injections were included as a blank run after each sample to avoid the carry-over effect. Data were acquired using "LabSolutions Insight" LC-MS software version 3.7 SP3 (workstation). Ionization was applied in both the positive and negative ion polarity modes in this study. At the same time, the data were collected

in the profile and centroid modes with a data storage threshold of 5000 absorbance for MS. The operating conditions were as follows: a detector voltage of 1.8 kV, a conversion dynode voltage of 10 kV, an interface voltage of 3 kV, an interface temperature of 300 °C, a desolvation line temperature of 250 °C, a heat block temperature of 400 °C, nebulizing gas argon at a flow of 3 L min$^{-1}$, heating gas carbon dioxide at a flow of 10 L min$^{-1}$, and drying gas nitrogen at a flow of 10 L min$^{-1}$. All phenolic compounds were observed in the programmed and optimized multiple reaction monitoring (MRM) mode (Supplementary Table S3). The representative chromatographic separation of the phenolic compounds in the four *V. opulus* extracts is depicted in Supplementary Figures S5–S9.

### 2.11. In Vitro Susceptibility Tests

2.11.1. The Agar Well Diffusion Method

The agar well diffusion method was used to investigate the antimicrobial potential of the *V. opulus* extracts against 19 test microorganisms (Table 1). For this purpose, wells with a diameter of 6 mm were punched in the Mueller Hilton broth (MHB, Ref. 102615973, Biolife, Milan, Italy) used for bacteria. Potato dextrose broth (PDB, Ref. 405172, Condalab, Laboratorios Canda S. A., Madrid, Spain) was used for the fungi supplemented with Difco™ bacteriological agar (Ref. 214530, Becton Dickinson and Company, Sparks, MD, USA). Afterward, 2 mL of saline containing a test microorganism with a turbidity of 0.5 McFarland units corresponding to $10^8$ CFU mL$^{-1}$ was prepared using a "DEN-1B" McFarland tube densitometer (Grant Instruments Ltd., Cambridge, United Kingdom). With the sterile swab, a microorganism inoculum was introduced into the agar surface, and the prepared well was finally filled with 0.15 mL of *V. opulus* extract. The zone of inhibition was observed and measured in mm, including the well diameter, after incubation for 24 h at $36 \pm 1$ °C.

**Table 1.** The reference test cultures, as well as their cultivation and assay media, used in the study.

| Microbial Test Culture | Collection Number | Assay | Cultivation and Re-Suspending |
|---|---|---|---|
| *Aspergillus brasiliensis* | ATCC 16404 | AWD | Potato Dextrose broth (PDB, Condalab, Laboratorios Canda S. A., Madrid, Spain, 405172); PDB supplemented with Difco™ Bacteriological agar (Becton Dickinson and Company, Sparks, MD, USA, Ref. 214530). |
| *Candida albicans* | ATCC 10231 | AWD | |
| *Bacillus cereus* | ATCC 11778 | AWD | |
| *Bacillus subtilis* | ATCC 6633 | AWD/MIC | |
| *Citrobacter freundii* | ATCC 43864 | AWD/ | |
| *Clostridium perfringens* | ATCC 13124 | AWD/ | |
| *Cronobacter muytjensii* | ATCC 51329 | AWD/MIC | |
| *Cronobacter sakazakii* | ATCC 29544 | AWD/ | Mueller Hinton Broth II (MHB, Merck Millipore Ltd., Cork, Ireland, Ref. 90922); MHB supplemented with Difco™ Bacteriological agar. |
| *Enterobacter cloacae* | ATCC 13047 | AWD/ | |
| *Enterococcus faecalis* | ATCC 29212 | AWD/MIC | |
| *Escherichia coli* | ATCC 8739 | AWD | |
| *Listeria innocua* | ATCC 33090 | AWD/MIC | |
| *Listeria ivanovii* | ATCC 33090 | AWD/MIC | |
| *Listeria monocytogenes* | ATCC 19112 | AWD | |
| *Pseudomonas aeruginosa* | ATCC 9027 | AWD/MIC | |
| *Salmonella enteritidis* | ATCC 13076 | AWD/MIC | |
| *Salmonella typhimurium* | ATCC 14028 | AWD/MIC | |
| *Staphylococcus aureus* | ATCC 6538P | AWD/MIC | |
| *Staphylococcus saprophyticus* | ATCC 15305 | AWD/MIC | |

Note: AWD and MIC—agar well diffusion or minimum inhibitory concentration methods, respectively. These methods were used to probe the antimicrobial activity of the *Viburnum opulus* L. extracts.

### 2.11.2. Minimum Inhibitory Concentration (MIC)

The minimum inhibitory concentration (MIC) of each *V. opulus* extract was determined using the microdilution method in 96-well plates and by utilizing ten reference test cultures, which were selected based on the results obtained during the agar well diffusion assay (Table 1). The analysis was conducted following the protocol reported by Radenkovs et al. [36] with modifications. Before the analysis, each test culture was re-suspended in MHB, and the suspension was adjusted to a final turbidity of 0.5 McFarland units. The 0.5 McFarland units corresponded to $10^8$ CFU mL$^{-1}$ of the test microorganism. An aliquot of 0.15 mL of each *V. opulus* extract and 0.075 mL of a 25% DMSO solution (DMSO:H$_2$O ratio 25:75 *v/v*) was introduced into the first well, and this amount was defined as the highest concentration of *V. opulus* extract, which corresponded to an absolute concentration in the well at 0.5 mL mL$^{-1}$. Before introducing the extract to the first well, it was filtered through a sterile 0.20 μm H-PTFE membrane filter to remove the non-dissolved plant fragments. The contents of the well were thoroughly mixed by dispensing the solution from the well several times with an automatic sterile pipette. Afterward, a serial two-fold dilution was used to obtain a 0.07 μL of the extract concentration, and this amount was defined as the lowest concentration of *V. opulus* extract, which corresponded to an absolute concentration in the final well at 0.24 μL mL$^{-1}$. When all the wells were filled with active material, 0.093 mL of 25% DMSO and 0.093 mL of MHB containing the test culture were added, thus reaching the final volume of 0.3 mL in each well. The viability of the bacterial cell was visualized using 10 μL per well of a 0.01% resazurin aqueous solution. The plates are incubated at 36 ± 1 °C for 24 h. The minimum inhibitory concentration (MIC) value was defined as the lowest concentration at which no viable microorganism growth was observed. A broad-spectrum amoxicillin/clavulanic acid antibiotic was used as a positive control. The MIC to amoxicillin/clavulanic acid was 0.004/0.0007 mg mL$^{-1}$ in all experiments. The MIC analysis was repeated to establish the influence of DMSO concentration on the growth of the microorganisms (the susceptibility of the microorganisms to DMSO was not observed at a DMSO range of 5 to 50%).

### 2.12. The Antioxidant Activity and Ferric Reducing Power of Extracts

#### 2.12.1. The DPPH$^\bullet$ Free Radical Scavenging Activity

The DPPH$^\bullet$ free radical scavenging activity of the *V. opulus* extracts was determined based on the methodology described by Radenkovs et al. [37] with slight modifications. Briefly, 0.15 mL of the above extract was mixed with 2.85 mL of DPPH$^\bullet$-EtOH solution (0.039 g DPPH$^\bullet$ in 1 L MeOH). The reaction proceeded at 22 ± 1 °C for 30 min in the dark. The absorbance of the extracts was measured at 0 and 30 min at a wavelength of 517 nm. The DPPH$^\bullet$ scavenging activity was expressed as the mg Trolox equivalent (TE) antioxidant capacity per 100 mL$^{-1}$ on a fresh weight basis (mg TE 100 mL$^{-1}$ FW) (Supplementary Table S1).

#### 2.12.2. Ferric-Reducing Antioxidant Power (FRAP)

The FRAP-reducing antioxidant power of the *V. opulus* extracts was determined using the procedure of Radenkovs et al. [38]. The fresh FRAP reagent was prepared using 300 mL of 0.3 M acetate buffer, a TPTZ solution in 40 mM of L$^{-1}$ HCl, and FeCl$_3$·6H$_2$O (20 mM L$^{-1}$). The three solutions were mixed at 10:1:1 (*v/v/v*) and then allowed to react at 37 ± 1 °C. The extracts and standard (FeS-O$_4$·7H$_2$O), or UPW, for blank (0.1 mL) were mixed with 3.6 mL of FRAP reagent, which was followed by incubation for 10 min in the dark at ambient temperature. The absorbance was measured at a wavelength of 593 nm. The FRAP values were expressed as a mg TE of ferric-reducing antioxidant power per 100 mL$^{-1}$ on a fresh weight basis (mg TE 100 mL$^{-1}$ FW) (Supplementary Table S1).

#### 2.12.3. The ABTS$^{\bullet+}$ Radical Cation Scavenging Activity

The ABTS$^{\bullet+}$ radical scavenging activity of the *V. opulus* extracts was determined according to the method of Balciunaitiene et al. [39] with slight modifications. The synthetic

radical ABTS$^{\bullet+}$ was prepared by mixing 0.15 mL of the 7.4 mM ABTS$^{\bullet+}$ solution with 0.15 mL of 2.6 mM potassium persulfate ($K_2S_2O_8$). The prepared solution was allowed to react for 16 h at room temperature, i.e., $22 \pm 1$ °C, in the dark. The ABTS$^{\bullet+}$ solution was then diluted with phosphate-buffered saline (PBS) with an adjusted pH of 7.4 to an absorbance of $0.70 \pm 0.02$ at 734 nm. An aliquot of 0.15 mL for each extract, standard, or blank (PBS) was added to 2.85 mL of diluted ABTS$^{\bullet+}$, which was then left to react for 10 min at $22 \pm 1$ °C in the dark. The absorbance of the extracts was measured at 734 nm. The ABTS$^{\bullet+}$ values were expressed as a mg TE of the antioxidant capacity per 100 mL$^{-1}$ on a fresh weight basis (mg TE 100 mL$^{-1}$ FW) (Supplementary Table S1).

*2.13. Statistical Analysis*

The results obtained are shown as the means ± standard deviation of the five replicates ($n = 5$) for the primer groups of compounds (total phenolics, flavonoids, tannins, and anthocyanins) and two replicates ($n = 2$) for the individual sugars, amino acids, and phenolics. A $p$-value of $\leq 0.05$ was used to denote the significant differences between the mean values; these differences were determined using a one-way analysis of variance (ANOVA) and Duncan's multiple range test, which were performed using "IBM® SPSS® Statistics" version 20.0 (SPSS Inc., Chicago, IL, USA). Pearson's correlation coefficient ($r$) was used to establish the correlation between the parameters investigated. A $p < 0.05$ value was adopted as the criteria for significant correlation.

## 3. Results

*3.1. The Individual Free Saccharide Content*

The results of the quantitative analysis of the free mono- and disaccharides in the *V. opulus* extracts are shown in Table 2. Generally, four individual monosaccharides were detected in the extracts of *V. opulus*, except in the flowers. Fructose was the primary sugar representative, fluctuating the concentration from 64.5 to 1132.8 mg 100 mL$^{-1}$ FW. The highest amount of fructose was observed in the *V. opulus* extract obtained from berries, whereas the lowest was in the bark extract. Glucose was the second most prevalent monosaccharide in the *V. opulus* extracts, with values ranging from 29.5 to 1103.1 mg 100 mL$^{-1}$ FW; the extract obtained from berries had the highest glucose content, while the extract from bark had the lowest.

**Table 2.** The concentration of the individual free saccharides in the *Viburnum opulus* L. extracts, in mg 100 mL$^{-1}$, FW.

| Component | Morphological Part | | | |
|---|---|---|---|---|
| | **Flowers** | **Mix** | **Bark** | **Berries** |
| Ribose | n.d. | $6.4 \pm 0.7$ [c] | $9.0 \pm 0.1$ [b] | $20.1 \pm 8.0$ [a] |
| Xylose | n.d. | $7.6 \pm 0.2$ | n.d. | n.d. |
| Fructose | n.d. | $348.3 \pm 11.9$ [b] | $64.5 \pm 1.6$ [c] | $1132.8 \pm 6.2$ [a] |
| Glucose | n.d. | $202.0 \pm 7.6$ [b] | $29.5 \pm 0.9$ [c] | $1103.1 \pm 6.0$ [a] |
| Sucrose | n.d. | n.d. | n.d. | n.d. |
| $\sum_{\text{Sugars}}$ | - | $564.4 \pm 20.4$ [b] | $103.0 \pm 2.6$ [c] | $2256.0 \pm 20.2$ [a] |

Note: Values are the means ± SD of the duplicates ($n = 2$). FW—the concentration expressed on a fresh weight basis; n.d.—not detected; and $\sum_{\text{Sugars}}$—sum of the total individual sugars. Means within the same saccharide with different superscript letters ([a–c]) represent significant differences, which are denoted with values at $p < 0.05$.

Ribose was also detected in the *V. opulus* extracts as a third monosaccharide, with the content ranging from 6.4 to 20.1 mg 100 mL$^{-1}$ FW. The highest value was observed in the berry-derived extract, while the lowest was found in the mixed *V. opulus* parts extract. Another interesting fact was the presence of free xylose in the mixed *V. opulus* parts extract. The *V. opulus* extracts were found to have varying amounts of total individual sugars, which ranged from 103.0 to 2256.0 mg 100 mL$^{-1}$ FW. The highest concentration was

observed in the *V. opulus* berry-derived extract, and the lowest concentration was found in the bark-derived extract.

### 3.2. The Individual Free Amino Acid Content

Based on the findings presented in Table 3, the *V. opulus* extracts contained high proline levels. The flowers and the mixed morphological parts (Mix) extract had the highest proline content, accounting for 63.3% and 36.2% of the total amino acids (AAs) and corresponding to 616.7 and 303.5 mg 100 mL$^{-1}$, respectively (Supplementary Figures S1 and S2). On the other hand, the proline concentration in the *V. opulus* extracts obtained from bark and berries was 58.6 and 15.5 mg 100 mL$^{-1}$, respectively.

**Table 3.** The concentration of the individual amino acids in the *Viburnum opulus* L. extracts, in mg 100 mL$^{-1}$, FW.

| Component | Morphological Part | | | |
|---|---|---|---|---|
| | **Flowers** | **Mix** | **Bark** | **Berries** |
| Cystine | 3.2 ± 0.2 [a] | 3.2 ± 0.0 [a] | 3.4 ± 0.4 [a] | 3.2 ± 0.0 [a] |
| Serine | 26.1 ± 0.8 [c] | 38.9 ± 0.2 [b] | 10.9 ± 0.7 [d] | 45.9 ± 5.4 [a] |
| Aspartic acid | 9.8 ± 0.2 [d] | 64.5 ± 1.0 [b] | 41.3 ± 7.8 [c] | 130.8 ± 10.2 [a] |
| Glycine | 15.1 ± 0.4 [a] | 6.9 ± 1.8 [b] | 3.2 ± 0.2 [c] | 3.4 ± 0.2 [c] |
| Threonine | 25.9 ± 0.3 [c] | 28.8 ± 1.1 [bc] | 13.6 ± 2.6 [d] | 32.2 ± 1.9 [a] |
| Glutamic acid | 50.2 ± 1.8 [d] | 103.6 ± 0.6 [b] | 81.0 ± 10.7 [c] | 156.6 ± 5.1 [a] |
| Histidine | 4.6 ± 0.1 [c] | 9.7 ± 0.1 [b] | BLQ | 14.2 ± 0.4 [a] |
| Alanine | 134.5 ± 1.0 [a] | 100.7 ± 0.9 [b] | 59.3 ± 5.2 [c] | 29.7 ± 0.2 [d] |
| Proline | 616.7 ± 1.5 [a] | 303.5 ± 0.5 [b] | 58.6 ± 6.0 [c] | 15.5 ± 0.3 [d] |
| Arginine | 43.6 ± 0.3 [bc] | 69.4 ± 1.0 [a] | 40.2 ± 1.5 [c] | 47.3 ± 0.1 [b] |
| Lysine | 4.5 ± 0.0 [b] | 16.7 ± 0.3 [a] | 4.5 ± 0.1 [b] | 17.9 ± 0.6 [a] |
| Valine | 13.9 ± 0.2 [b] | 18.0 ± 0.4 [a] | 9.0 ± 0.4 [bc] | 7.9 ± 0.4 [c] |
| Methionine | 0.8 ± 0.0 [b] | 1.7 ± 0.2 [ab] | BLQ | 2.9 ± 0.1 [a] |
| Tyrosine | 14.6 ± 0.3 [a] | 12.7 ± 0.8 [a] | 2.8 ± 0.3 [c] | 6.6 ± 0.1 [b] |
| Leucine | 2.4 ± 0.3 [c] | 30.0 ± 2.5 [a] | 9.5 ± 0.3 [b] | 9.5 ± 0.3 [b] |
| Isoleucine | 5.1 ± 0.0 [c] | 19.2 ± 0.0 [a] | 9.3 ± 0.4 [b] | 4.9 ± 0.2 [c] |
| Phenylalanine | 3.1 ± 0.0 [c] | 10.5 ± 0.2 [a] | 5.8 ± 0.2 [bc] | 8.4 ± 0.1 [a] |
| $\sum_{EAAs}$ | 60.3 ± 1.0 [c] | 134.5 ± 4.8 [a] | 51.6 ± 3.9 [d] | 98.0 ± 4.0 [b] |
| $\sum_{BCAAs}$ | 21.3 ± 0.5 [c] | 67.2 ± 3.0 [a] | 27.8 ± 1.0 [b] | 22.3 ± 0.9 [c] |
| $\sum_{Total}$ | 974.1 ± 7.4 [a] | 838.1 ± 11.5 [b] | 352.4 ± 36.7 [d] | 537.0 ± 25.7 [c] |

Note: Values are the means ± SD of the duplicates (*n* = 2). FW—the concentration expressed on a fresh weight basis; BLQ—the observed concentration is below the limit of quantification; $\sum_{EAAs}$—sum of the essential amino acids (i.e., histidine, isoleucine, leucine, lysine, methionine, phenylalanine, threonine, and valine); $\sum_{BCAAs}$—sum of the branched-chain amino acids (i.e., leucine, isoleucine, and valine); and $\sum_{Total}$—sum of the total individual amino acids. Means within the same amino acid with different superscript letters ([a–d]) represent significant differences, which are denoted with values at *p* < 0.05.

The glutamic acid content was the second most prevalent AA identified in the *V. opulus* extracts, and it corresponded to 50.2 to 156.6 mg 100 mL$^{-1}$. The highest glutamic acid content was found in the berry extract, while the lowest was found in the flower extract. The extract derived from a mixture of *V. opulus* morphological parts also demonstrated a remarkable amount of glutamic acid due to the presence of berries in the extraction mixture, the value of which corresponded to 103.6 mg 100 mL$^{-1}$. Apart from glutamic acid, the *V. opulus* extracts also demonstrated a relative abundance of aspartic acid, which contributed to a total AA amount range of 1.0% to 24.4%, with the berry extract having the highest amount and flower extract having the lowest. The concentrations of the EAAs and BCAAs in the *V. opulus* extracts varied from 51.6 to 134.5 mg 100 mL$^{-1}$ FW and 21.3 to 67.2 mg 100 mL$^{-1}$ FW, respectively. The content of the total individual AAs in the *V. opulus* extracts varied from 352.4 to 974.1 mg 100 mL$^{-1}$, with the extracts obtained from flowers having the highest level and the extracts obtained from bark the lowest.

### 3.3. Total Phenolic Content

The concentration of the TPC in the *V. opulus* extracts ranged from 68.7 to 379.5 mg GAE 100 mL$^{-1}$ FW, with the flower extract having a statistically ($p < 0.05$) lower value and the berry extract having a higher value (Table 4).

**Table 4.** The concentration of the group compounds in the *Viburnum opulus* L. extracts, in mg 100 mL$^{-1}$, FW.

| Compound | Morphological Part | | | |
|---|---|---|---|---|
| | **Flowers** | **Mix** | **Bark** | **Berries** |
| TPC | 68.7 ± 0.9 [d] | 261.6 ± 0.7 [c] | 323.6 ± 1.8 [b] | 379.5 ± 1.6 [a] |
| TFC | 11.1 ± 0.4 [d] | 79.6 ± 0.9 [c] | 96.4 ± 3.6 [b] | 130.4 ± 0.7 [a] |
| TTC | 46.9 ± 0.3 [d] | 228.6 ± 1.3 [c] | 313.5 ± 3.6 [b] | 325.3 ± 3.4 [a] |
| TAC | n.d. | 3.0 ± 0.1 [b] | n.d. | 14.7 ± 0.2 [a] |

Note: Values are the means ± SD of the quintuplicates (*n* = 5). TPC—total phenolic content; TFC—total flavonoid content; TTC—total tannin content; TAC—total anthocyanin content; FW—the concentration expressed on a fresh weight basis; and n.d.—not detected. The means within the same compound with different superscript letters [a–d] represent significant differences, which are denoted with values at $p < 0.05$.

### 3.4. Total Flavonoid Content

As in the case of TPC, the highest total flavonoid content (TFC) was found in the *V. opulus* extract obtained from berries, and the lowest was from flowers, which corresponded to 130.4 and 11.1 mg CE 100 mL$^{-1}$ FW, respectively (Table 4).

### 3.5. Total Tannin Content

As part of the further research on the extracts obtained from *V. opulus*, an abundance of TTC was revealed. The concentration of TTC fluctuated from 46.9 to 325.3 mg CE 100 mL$^{-1}$ FW, with the flower extract having a statistically ($p < 0.05$) lower value and the berry and bark extracts having higher ones (Table 4).

### 3.6. Total Anthocyanin Content

Further spectrophotometric analysis was conducted to study the TAC of the *V. opulus* extracts, which confirmed that the presence of such pigments was in only two of the four *V. opulus* extracts, i.e., the mixed morphological part (Mix) extract and the berry extract. The TAC concentration was found to be between 3.0 and 14.7 mg CGE 100 mL$^{-1}$ FW, with the mix extract having a statistically ($p < 0.05$) lower value and the berry extract having a higher value (Table 4).

### 3.7. Individual Phenolic Content

According to the results, chlorogenic acid was the most prevailing compound identified in the *V. opulus* extracts as it fluctuated from 5.69 to 149.64 mg 100 mL$^{-1}$ FW (Table 5). Also, other quinic acid derivatives, including neochlorogenic and cryptochlorogenic acids, were found in the extracts of *V. opulus*. However, the latter was represented exclusively in the extract derived from bark. The concentration of these hydroxycinnamates varied from 0.05 to 4.75 mg 100 mL$^{-1}$ FW, with the bark extract having the highest content and flower and berry extracts having the lowest. The catechin concentration in the bark extract was 24.82 mg 100 mL$^{-1}$ FW, while it was 12.85 mg 100 mL$^{-1}$ FW in the berry extract. The presence of luteolin as the third most prevailing phenolic was observed in the extracts of *V. opulus* from 0.01 to 1.15 mg 100 mL$^{-1}$ FW, with the extract derived from flowers having the highest amount and the extract derived from berries and bark having the lowest.

**Table 5.** The concentration of the individual phenolics in the *Viburnum opulus* L. extracts, in mg 100 mL$^{-1}$, FW.

| Component | Morphological Part | | | |
|---|---|---|---|---|
| | **Flowers** | **Mix** | **Bark** | **Berries** |
| Vanillin | 0.02 ± 0.00 [a] | 0.02 ± 0.00 [a] | 0.02 ± 0.00 [a] | 0.03 ± 0.00 [a] |
| Quercetin | 0.04 ± 0.00 [c] | 0.05 ± 0.00 [bc] | 0.03 ± 0.00 [b] | 0.09 ± 0.00 [a] |
| Isorhamnetin | 0.04 ± 0.00 [a] | n.d. | 0.01 ± 0.00 [b] | 0.01 ± 0.00 [b] |
| Kaempferol | 0.10 ± 0.01 [a] | 0.05 ± 0.02 [b] | BLQ | BLQ |
| Gallic acid | n.d. | n.d. | BLQ | BLQ |
| Neochlorogenic acid | 0.29 ± 0.02 [c] | 1.68 ± 0.01 [b] | 4.75 ± 0.17 [a] | 0.32 ± 0.00 [c] |
| Protocatechuic acid | 0.02 ± 0.00 [c] | 0.10 ± 0.00 [ab] | 0.13 ± 0.00 [a] | 0.07 ± 0.00 [b] |
| Chlorogenic acid | 5.69 ± 0.12 [d] | 49.36 ± 0.02 [b] | 11.07 ± 0.15 [c] | 149.64 ± 0.13 [a] |
| Cryptochlorogenic acid | n.d. | n.d. | 0.05 ± 0.00 | n.d. |
| (±)-Catechin | BLQ | 13.74 ± 0.15 [b] | 24.82 ± 0.15 [a] | 12.85 ± 0.20 [c] |
| (−)-Epicatechin | BLQ | 1.09 ± 0.01 [b] | 1.65 ± 0.03 [a] | 1.60 ± 0.10 [a] |
| Caffeic acid | 0.07 ± 0.01 [d] | 0.16 ± 0.01 [bc] | 0.29 ± 0.00 [a] | 0.12 ± 0.1 [c] |
| Rutin | 0.32 ± 0.09 [b] | 0.34 ± 0.03 [b] | 0.08 ± 0.00 [c] | 0.77 ± 0.03 [a] |
| *trans*-Ferulic acid | n.d. | n.d. | n.d. | 0.12 ± 0.53 |
| *para*-Coumaric acid | 0.34 ± 0.05 [a] | 0.16 ± 0.03 [b] | 0.07 ± 0.01 [c] | 0.07 ± 0.00 [c] |
| Luteolin | 1.15 ± 0.02 [a] | 0.21 ± 0.02 [b] | 0.01 ± 0.00 [c] | 0.01 ± 0.00 [c] |
| Luteolin-7-*O*-glucoside | n.d. | n.d. | n.d. | 0.01 ± 0.00 [c] |
| Rhamnetin | 0.02 ± 0.00 [a] | 0.01 ± 0.00 [a] | 0.003 ± 0.00 [b] | 0.0004 ± 0.00 [b] |
| $\sum$Total | 8.11 ± 0.32 [d] | 66.95 ± 0.29 [b] | 42.98 ± 0.51 [c] | 165.71 ± 1.00 [a] |

Note: Values are the means ± SD of the duplicates (*n* = 2). FW—the concentration expressed on a fresh weight basis; BLQ—the observed concentration is below the limit of quantification; n.d.—not detected; and $\sum$Total—sum of the total individual phenolic compounds. The means within the same phenolic compound with different superscript letters ([a–d]) represent significant differences, which are denoted with values at $p < 0.05$.

### 3.8. Antioxidant Activity

The DPPH$^\bullet$ radical scavenging activity of the *V. opulus* extracts depicted in Figure 3 showed values ranging between 54.8 and 256.2 mg TE 100 g$^{-1}$ FW. Moreover, the AOA results of the *V. opulus* extracts that were determined by the FRAP method were similar to those obtained by DPPH$^\bullet$, i.e., they varied from 30.7 to 249.3 mg TE 100 g$^{-1}$ FW. It is worth noting that the values obtained by ABTS$^{\bullet+}$ were two-fold higher than those observed by the DPPH$^\bullet$ and FRAP methods, even though the contribution order of the *V. opulus* morphological parts to AOA remained intact. The calculated Pearson correlation coefficients (Table 6) indicated that the most remarkable contribution of the group compounds to AOA was attributed to the TTC and TFC.

However, a moderate correlation between the AOA values determined by the DPPH$^\bullet$, FRAP, and ABTS$^{\bullet+}$ methods and the content of the chlorogenic acid in the *V. opulus* berry and mix extracts was observed. This was because the Pearson correlation coefficients for them were found as follows: $r = 0.5501$, $r = 0.5954$, and $r = 0.5944$, respectively. A strong correlation was found between individual phenolics such as catechin and the AOA values, which were estimated by the DPPH$^\bullet$, FRAP, and ABTS$^{\bullet+}$ methods and corresponded to $r = 0.8527$, $r = 0.8321$, and $r = 0.8324$, respectively. Due to the remarkably denser bioactive composition, as spectrophotometric and chromatographic approaches revealed, the extracts from the *V. opulus* berries and bark demonstrated statistically better ($p < 0.05$) AOA values than those obtained from the mixed (Mix) and flower extracts.

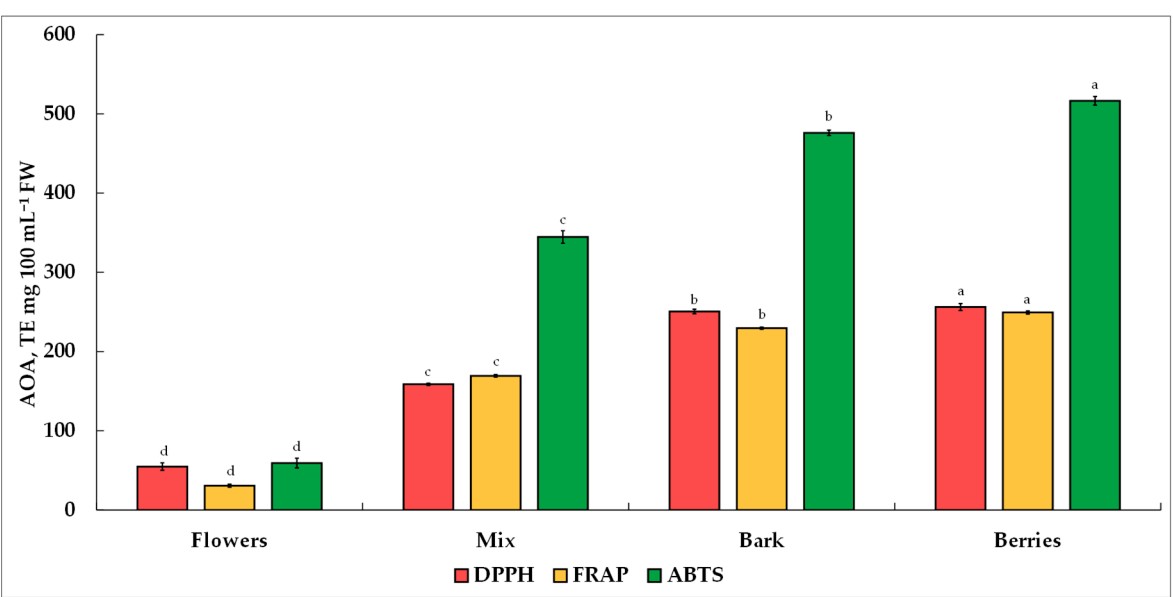

**Figure 3.** Antioxidant activity of the *Viburnum opulus* L.-derived extracts according to certain determination methods. Note: Values are the means ± SD of the quintuplicates (*n* = 5). The means within the same antioxidant activity screening method with different superscript letters ($^{a-d}$) represent significant differences, which are denoted with values at *p* < 0.05. FW—the antioxidant activity expressed on a fresh weight basis.

**Table 6.** The correlations between the phenolics, flavonoids, tannins, and individual amino acids, as well as the individual phenolic compounds and antioxidant activity, as determined by the DPPH$^{\bullet+}$, FRAP, and ABTS$^{\bullet+}$ methods.

| Variable | DPPH | FRAP | ABTS | TIS | TIAA | TPC | TFC | TTC | TAC | TIPC | CGA | CT | CA | RT |
|---|---|---|---|---|---|---|---|---|---|---|---|---|---|---|
| DPPH | 1 | | | | | | | | | | | | | |
| FRAP | 0.9938 | 1 | | | | | | | | | | | | |
| ABTS | 0.9950 | 0.9999 | 1 | | | | | | | | | | | |
| TIS | 0.5597 | 0.5988 | 0.5985 | 1 | | | | | | | | | | |
| TIAA | −0.9407 | −0.9014 | −0.9065 | −0.5257 | 1 | | | | | | | | | |
| TPC | 0.9830 | 0.9960 | 0.9955 | 0.6639 | −0.8833 | 1 | | | | | | | | |
| TFC | 0.9625 | 0.9804 | 0.9799 | 0.7430 | −0.8667 | 0.9938 | 1 | | | | | | | |
| TTC | 0.9951 | 0.9990 | 0.9990 | 0.5627 | −0.9026 | 0.9912 | 0.9707 | 1 | | | | | | |
| TAC | 0.5270 | 0.5642 | 0.5643 | 0.9987 | −0.5057 | 0.6308 | 0.7130 | 0.5272 | 1 | | | | | |
| TIPC | 0.6891 | 0.7291 | 0.7282 | 0.9831 | −0.6226 | 0.7853 | 0.8493 | 0.6979 | 0.9731 | 1 | | | | |
| CGA | 0.5501 | 0.5954 | 0.5944 | 0.9981 | −0.4963 | 0.6624 | 0.7416 | 0.5591 | 0.9954 | 0.9840 | 1 | | | |
| CT | 0.8527 | 0.8321 | 0.8324 | 0.0542 | −0.7650 | 0.7817 | 0.7075 | 0.8559 | 0.0123 | 0.2277 | 0.0506 | 1 | | |
| CA | 0.6545 | 0.6133 | 0.6147 | −0.2601 | −0.6105 | 0.5417 | 0.4456 | 0.6477 | −0.2976 | −0.0927 | −0.2685 | 0.9459 | 1 | |
| RT | −0.8405 | −0.7843 | −0.7917 | −0.5054 | 0.9739 | −0.7664 | −0.7582 | −0.7834 | −0.4960 | −0.5700 | −0.4651 | −0.6352 | −0.5088 | 1 |

Note: Pearson's correlation coefficient (*r*) was set as significant at the *p* ≤ 0.05 level. TPC, TFC, TTC, and TAC—the content of phenolics, flavonoids, tannins, and anthocyanins, respectively; TIS, TIAA, and TIPC—the content of the total individual saccharides, amino acids, and phenolics, respectively; and CGA, CT, CA, and RT—the content of the chlorogenic acid, catechin, caffeic acid, and rutin, respectively.

### 3.9. The Antimicrobial Activity According to the Inhibition Zone Values

As part of determining the antimicrobial activity of *V. opulus*, the *V. opulus* extracts derived from berries demonstrated remarkable activity against Gram-negative *C. muytjensii* and *P. aeruginosa* bacteria by producing inhibition zones of 28.6 and 22.4 mm, respectively (Table 7). Among the other extracts investigated, the *V. opulus* extract derived from berries exhibited the most remarkable antimicrobial activity against Gram-positive bacteria *S. aureus* by producing a zone of inhibition of 28.1 mm.

**Table 7.** The antibacterial activity of the *Viburnum opulus* L. extracts against the selected Gram-negative and Gram-positive bacteria test cultures, which were assessed according to their diameter of inhibition zone values.

| Test Culture | | Average Zone of Inhibition, mm | | | | |
|---|---|---|---|---|---|---|
| | Gram Type | Flowers | Mix | Bark | Berries | Amoxicillin/Clavulanic Acid * |
| *B. subtilis* | Gram-positive | - | $15.4 \pm 0.1$ [a] | $13.1 \pm 0.2$ [b] | $11.2 \pm 0.3$ [c] | |
| *S. aureus* | | - | $13.1 \pm 0.1$ [b] | $12.1 \pm 0.1$ [b] | $28.1 \pm 0.3$ [a] | |
| *S. saprophyticus* | | - | $20.2 \pm 0.2$ [a] | $14.1 \pm 0.1$ [b] | $15.4 \pm 0.4$ [b] | |
| *L. innocua* | | $23.1 \pm 0.1$ | - | - | - | $22.5-39.3$ |
| *L. ivanovii* | | - | - | - | $12.2 \pm 0.2$ | |
| *E. faecalis* | | - | $12.3 \pm 0.1$ [c] | $14.3 \pm 0.3$ [b] | $19.4 \pm 0.2$ [a] | |
| *S. enteritidis* | Gram-negative | - | $12.7 \pm 0.1$ [b] | $14.5 \pm 0.2$ [a] | - | |
| *S. typhimurium* | | - | - | $10.2 \pm 0.1$ [b] | $12.0 \pm 0.1$ [a] | |
| *C. muytjensii* | | - | $16.2 \pm 0.2$ [c] | $20.4 \pm 0.1$ [b] | $28.6 \pm 0.1$ [a] | |
| *P. aeruginosa* | | - | - | $14.1 \pm 0.2$ [b] | $22.4 \pm 0.1$ [a] | |

Note: Values are expressed as the means $\pm$ SD values of the duplicates ($n = 2$). Means within the same test microorganism with different superscript letters ([a–c]) represent significant differences, which are denoted with values at $p \leq 0.05$. * The MIC of amoxicillin/clavulanic acid at a 500.0 mg mL$^{-1}$ concentration was used as a positive control.

The extracts obtained from the *V. opulus* morphological parts, except flowers and berries, did not show any antimicrobial activity against *L. innocua* and *L. ivanovii*.

### 3.10. The Antimicrobial Activity According to the Minimum Inhibitory Concentration Values

According to the results displayed in Table 8 and Figure 4, the extracts obtained from the mix, bark, and berries of *V. opulus* were more active against the selected bacteria at lower extract concentrations. These extracts' minimum inhibitory concentration (MIC) values were 0.49 for the mix and 0.24 µL mL$^{-1}$ for the bark and berry extracts. No inhibitory activity of the extracts obtained from a mix, bark, and berries of *V. opulus* was observed against the selected test cultures at high extract concentrations of 500.0 µL mL$^{-1}$. Similarly, the extract obtained from the flowers of *V. opulus* did not show any significant inhibitory activity against the selected bacteria, except for *L. innocua*, at either a high extract dose of 500 µL mL$^{-1}$ or a low extract dose of 0.24 µL mL$^{-1}$.

**Table 8.** The minimum inhibitory (MIC) concentration values of the *Viburnum opulus* L. extracts against the selected Gram-negative and Gram-positive bacteria test cultures.

| Test Culture | | Average MIC, µL mL$^{-1}$ | | | | |
|---|---|---|---|---|---|---|
| | Gram Type | Flowers | Mix | Bark | Berries | Amoxicillin/Clavulanic * Acid, mg mL$^{-1}$ |
| *B. subtilis* | Gram-positive | - | $31.2 \pm 0.0$ [a] | $0.24 \pm 0.0$ [b] | $0.24 \pm 0.0$ [b] | |
| *S. aureus* | | - | $0.24 \pm 0.0$ [a] | $0.24 \pm 0.0$ [a] | $0.24 \pm 0.0$ [a] | |
| *S. saprophyticus* | | - | $0.24 \pm 0.0$ [a] | $0.24 \pm 0.0$ [a] | $0.24 \pm 0.0$ [a] | |
| *L. innocua* | | $0.24 \pm 0.0$ | - | - | - | $0.0021/0.00007$ |
| *L. ivanovii* | | - | - | - | $0.24 \pm 0.0$ | |
| *E. faecalis* | | - | - | $0.24 \pm 0.0$ [a] | $0.24 \pm 0.0$ [a] | |
| *S. enteritidis* | Gram-negative | - | $0.49 \pm 0.0$ [a] | $0.24 \pm 0.0$ [b] | - | |
| *S. typhimurium* | | - | - | $0.24 \pm 0.0$ [b] | $0.24 \pm 0.0$ [b] | |
| *C. muytjensii* | | - | $0.24 \pm 0.0$ [a] | $0.24 \pm 0.0$ [a] | $0.24 \pm 0.0$ [a] | |
| *P. aeruginosa* | | - | - | $0.24 \pm 0.0$ [a] | $0.24 \pm 0.0$ [a] | |

Note: Values are expressed as the means $\pm$ SD values of the duplicates ($n = 2$). Means within the same test microorganism with different superscript letters ([a–b]) represent significant differences, which are denoted with values at $p \leq 0.05$. * The MICs of the amoxicillin/clavulanic acid at a concentration range of 4.37 to 0.002 and 0.007 to 0.00003 mg mL$^{-1}$ were used as the positive controls, respectively.

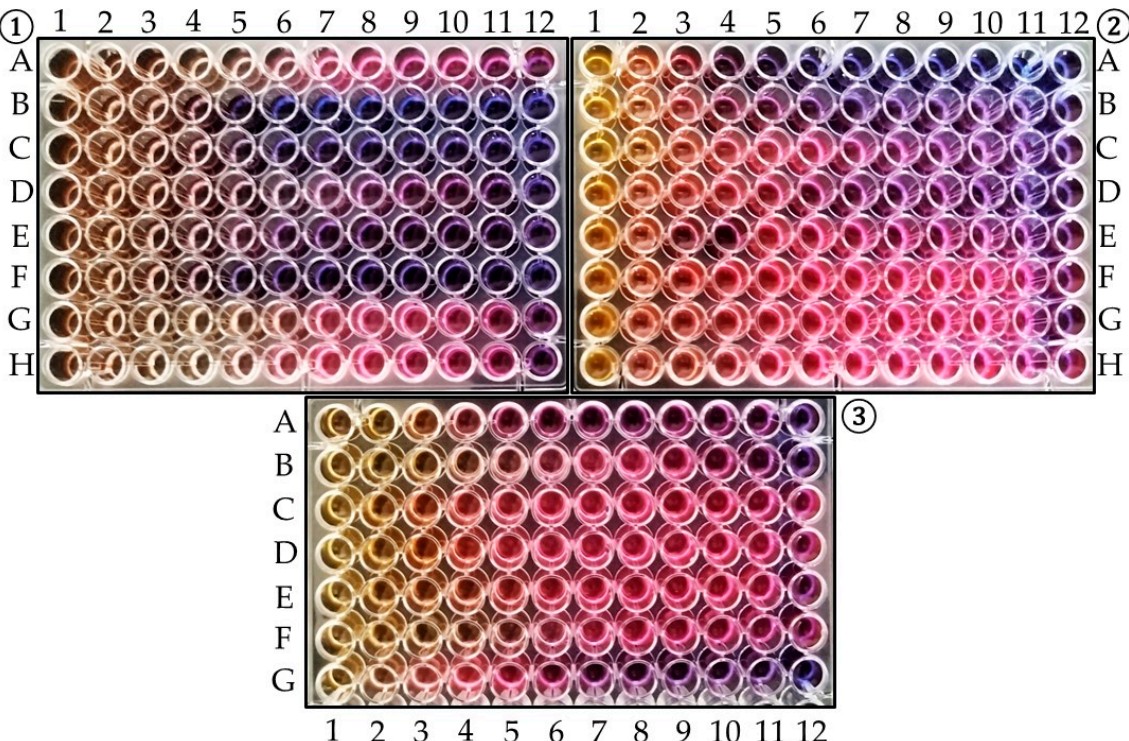

**Figure 4.** Determination of the minimum inhibitory concentration (MIC) using the microdilution method. Note: The pink color represents the growth of the microorganism, and the purple color represents no growth. Number 1 indicates the highest concentration and number 12 indicates the lowest concentration of antimicrobial agent in the wells. The cycled Digits 1 and 2 on the microplates represent the antimicrobial activity of the extracts derived from the berries and bark of *Viburnum opulus* L. Cycled Digit 3 on the microplate represents the antimicrobial activity of the extracts derived from all the morphological parts (Mix) (A3–F3) and the flowers (G3) of *Viburnum opulus* L. A1—*L. ivanovii*; B1—*P. aeruginosa*; C1—*C. muytjensii*; D1—*S. saprophyticus*; E1—*S. typhimurium*; F1—*B. subtilis*; G1—*E. faecalis*; H1—*S. aureus*; A2—*P. aeruginosa*; B2—*C. muytjensii*; C2—*S. saprophyticus*; D2—*S. typhimurium*; E2—*B. subtilis*; F2—*E. faecalis*; G2—*S. aureus*; H2—*S. enteritidis*; A3—*C. muytjensii*; B3—*S. saprophyticus*; C3—*B. subtilis*; D3—*E. faecalis*; E3—*S. aureus*; F3—*S. enteritidis*; and G3—*L. innocua*.

## 4. Discussion

The study conducted by Mizzi et al. [40] found that sugars can act as either synergists or antagonists to pathogenic microorganisms by either promoting or demoting their growth. Based on this finding, the first step in analyzing *V. opulus* extracts was establishing their sugar profile and content. Perova et al. [1] observed double to quadruple as much fructose in *V. opulus* berries, which fluctuated from 2650 to 4040 mg 100 $g^{-1}$. The observed ratio of glucose to fructose in the extract of a *V. opulus* berry was 1:1, which is consistent with the findings reported by Perova et al. [1] for *V. opulus* berries, who indicated a glucose content from 2860 to 4640 mg 100 $g^{-1}$. The presence of glucose in the fruits and flowers of *V. opulus* was previously reported by Polka et al. [18], who reported concentrations ranging from 2.0 to 15.3 g 100 $mL^{-1}$ on a dry weight basis (DW). However, those authors also reported an absence of this sugar in bark extract. The variation in fructose and glucose content can be attributed to the origin of the fruit, the type of morphological part, the extraction method used, and the expression of the results. This is the first report highlighting ribose's presence in *V. opulus* extracts. However, this sugar alcohol has already been documented in mulberry, apple, grape, apricot, and pear juice extracts [41]. Xylose, a hemicellulose-derived sugar that is usually extracted after partial or complete hydrolysis of lignocellulose materials [42], has also been observed as part of *V. opulus* extracts derived from a mixture of morphological parts. However, the presence of xylose, as the structural component of opulosides and

anthocyanidins, was also mentioned by Perova et al. [1], who detected it as part of the non-target profiling of iridoid glucosides and anthocyanins. A similar observation was made by Kajszczak et al. [7], who revealed the abundance of iridoids in green parts, mainly in the leaves and young stems, of the *V. opulus* plant. This observation reinforces the funding of the present study since the extract was prepared using all the morphological parts of the *V. opulus* shrub, i.e., flowers, whole berries, leaves, buds, and bark. Prolonged ultrasonic treatment promotes the release of structurally bound cell components, including xylose moieties, as was reported by Wang et al. [43]. The release of bound components is conditioned by an ultrasound's ability to increase cell wall permeability, and it will eventually disrupt it, thus leading to a more efficient mass transfer between the solvent and plant material. This statement can be reinforced by an early observation made by Sun et al. [44]. It is worth noting that the total sugar content in *V. opulus* extracts did not exceed 2.5%, which is far from the 20.0% concentration required to inhibit the growth of *S. aureus* and *E. coli*, as stated by Mizzi et al. [40]. On the other hand, sugars in small quantities can be regarded as growth-promoting compounds since they serve as substrates for the microorganism, resulting in enhanced microbial growth instead of inhibition [45].

Along with the undebatable physiological benefits inherent to amino acids (AAs) [46], which were long considered simply building blocks for protein synthesis [46], it has recently been shown that AAs act not only as growth-promoting substrates for pathogenic bacteria and opportunistic pathogens, but they also support the bacteria evasion that occurs in host immune defense [47]. Therefore, as in the case of saccharides, the profile and concentrations of individual free amino acids in *V. opulus* extracts were specified for the first time. The results make it possible to draw an interconnection between the AA content and the antimicrobial activity of *V. opulus* extracts. The AAs analysis conducted by HPLC-ESI-TQ-MS/MS revealed relative content and distribution patterns over four *V. opulus* extracts. The proline concentration in the *V. opulus* extracts obtained from bark and berries was four and twenty-one times lower than in flower-derived extract. A high proline concentration in flower-derived extract has been associated with its role in plant metabolism and resistance to multiple abiotic and biotic stresses during flowering and plant growth and development [48]. Hosseinifard et al. [49] highlighted the dominance of proline in tomato flowers, revealing a 60 times higher concentration than that observed in other morphological parts such as leaves, roots, and fruit. Despite the contribution of dietary proline to collagen biosynthesis [50], its direct involvement in the proliferation of disease-causing, biofilm-forming pathogenic microorganisms such as *S. aureus*, *E. coli*, *E. chaffeensis*, *Clostridium difficile*, *Helicobacter* spp., and *Cryptococcus neoformans* was highlighted by Christgen et al. [51] and Cleaver et al. [52], who revealed its role as a respiratory substrate and osmolyte that ensures protection against environmental stress. Glutamic acid is the second most prevalent amino acid (AA) identified in the *V. opulus* extracts. Its contribution to the total AA amount was 5.2 to 29.2% in the *V. opulus* extracts. The highest glutamic acid content was found in the berry extract, while the lowest was in the flower extract. The presence of berries in the extract derived from a mixture of morphological parts can reinforce glutamic acid's prevalence, as Asadpoor et al. [53] reported for the nine fruit juices. The role of glutamic acid as a sole carbon and nitrogen source for microorganisms was revealed by Liu et al. [54], who indicated a growth-promoting ability toward *S. enterica* and *Pseudomonas* spp. A recent study showed disparities in the AA metabolism in *C. albicans* biofilms, with a prominent upregulation of arginine, proline, aspartate, and glutamate metabolism in high biofilm-forming isolates [55]. The presence of alanine in the extracts of *V. opulus*, specifically in the flower extract, makes it attainable and worthwhile to assume a possible contribution of this extract to the growth of selected opportunistic bacteria. This is relevant as the extensive proliferation of *E. coli* K-12 and EC-14 was observed by Muranaka et al. [56] in infected model mice with an excess of alanine. This observation was supported by Díaz-Pascual et al. [57], who noted that *E. coli* colonies in a controllable biofilm model system metabolized alanine as a carbon source in oxic conditions rather than in anoxic ones. The high demand for alanine in bacteria is conditioned by its direct involvement in the

synthesis of peptidoglycan, a pivotal component of the cell wall of both Gram-negative and Gram-positive bacteria [58]. The observed EAA and BCAA values in the *V. opulus* extracts were much higher than those reported for the vinegar [59] and fermented juice [60] of *V. opulus*. The content of the total individual AAs in the extracts varied significantly (*p* < 0.05) between the investigated *V. opulus* morphological parts. However, the highest total AA content was found in the extract obtained from flowers of *V. opulus*. A similar observation was made by Kajszczak et al. [7], who indicated that the flowers of *V. opulus* are more abundant in protein than the fruit, stalk, and leaves. Overall, from a nutritional standpoint, the extracts obtained from flowers and a mix of *V. opulus* morphological parts can be highlighted as exceptional, as the AA levels were found to be many times higher than those reported by others for plant extracts. However, recent findings have indicated that some AAs can induce the growth of biofilm-forming bacteria, such as *E. coli*, *P. aeruginosa*, and *S. aureus*. Therefore, additional studies are required to determine the influence of *V. opulus* extracts on the growth of opportunistic bacteria.

Previous research has examined the chemical composition of different parts of *V. opulus*, including its fruits, flowers, and bark, indicating a relative abundance of bioactives and their fluctuations over different morphological parts [61]. However, most of these studies were based on investigating the dried morphological parts of the *V. opulus* plant rather than extracts in their original state. Therefore, it is essential to comprehend better the profile and concentrations of the bioactives present in *V. opulus* extracts. Preliminary spectrophotometric studies have shown that group compounds, particularly TPC, are present in ample amounts, as indicated in Table 4. The values observed in the present study contradict the findings of Polka and Podsędek [62], as well as Polka et al. [18], for dried *V. opulus*-obtained extracts and *V. opulus* parts, who highlighted the superiority of the *V. opulus* bark over both flowers and fruits. Dienaite et al. [63], studying the yield of TPC from dried *V. opulus* pomace as the function of extraction solvent utilized, observed relatively higher TPC values obtained by ethanol rather than water as a sole solvent. The concentrations of TPC in their study were four to fifteen times higher than in the present study. The choice of solvent should depend on the type of matrix and the nature of the substances being extracted. Plant materials contain hydrophilic compounds, i.e., compounds that are preferably soluble in water, but these do not appear in organic solvents and vice versa. In the research of Radenkovs et al. [37], an augmented effect upon extractability of TPC from wild crab apple pomace and higher antioxidant activity (AOA) was obtained by utilizing 30% ethanol instead of other solvents or their aqueous solutions. On this occasion, water played the swelling agent role, increasing the plant cell contact surface [64]. Since the studied *V. opulus* extracts were intended as a homeopathic remedy for the preventive treatment of colds in children and adults, water was used as the only solvent to prepare the extracts industrially. The high TPC content in the *V. opulus* extracts derived from berries and bark is expected to contribute to the inhibition of opportunistic microorganisms, as reported by Mahboubi et al. [65] and Ispiryan et al. [66] for *Punica granatum* L. and *Rubus idaeus* L. extracts, respectively.

The observed TFC values were partially similar to those reported by Düz et al. [67] for *V. opulus* extracts derived from berries, flowers, and branches when using four different solvents for extraction. The authors found the highest concentration of TFC in the ethyl acetate extracts of *V. opulus* obtained from leaves, and the lowest was found in the water extracts. However, Mahboubi et al. [65] highlighted a statistically better extractability of TFC from *Punica granatum* L. flowers by methanol, whereas those extracted by water and ethanol were much worse. Relatively lower values of TFC were observed in the *V. opulus* extracts of the current study, which were conditioned by the weak solubility of flavonoids in water compared with organic solvents such as acetonitrile, methanol, ethanol, or ethyl acetate [64]. Like in the case of TPC, a strong and positive correlation between TFC and the growth of pathogenic bacteria was revealed by Sartini et al. [68], who indicated their anti-staphylococcal activity toward suppressing the growth of *S. aureus*. In a detailed study on the antimicrobial activity of flavonoids, Donadio et al. [69] reported the possible

action mechanisms of flavonoids on the growth of pathogenic bacteria, among which the inhibition of transporters such as efflux pumps responsible for the extrusion of noxious compounds, both in Gram-positive and Gram-negative bacteria was highlighted. The synergistic effect of plant material-derived flavonoids on the cell wall integrity has also been revealed, indicating the potential contribution toward disruptions of membrane ion exchange and reducing ATP synthesis by depleting cell membranes. The relative ampleness of molecules containing substituents such as -OH in the B-ring defines the antimicrobial activity of a particular compound [70].

The observed TTC values partially agree with those reported by Polka and Podsędek [62], who indicated that the TTC in *V. opulus* extracts range from 327.0 to 5029 mg (cyanidin equivalents, CYE) 100 mL$^{-1}$ DW. Interestingly, the highest content of TFC was observed in bark-derived extracts rather than berry ones, as found in the current study. According to the comprehensive research by Ucella-Filho et al. [71], tannin-rich water extracts obtained from the plants of families such as *Fabaceae*, *Combretaceae*, *Anacardiaceae*, *Punicaceae*, and *Curtisiaceae* have been shown to have prominent antimicrobial activity against various types of bacteria, including *E. coli*, *L. innocua*, *S. aureus*, *E. faecalis*, *B. cereus*, *Shigella dysenteriae*, *P. aeruginosa*, *S. enterica*, *S. typhimurium*, *S. flexneri*, *Neisseria gonorrhoea*, *K. pneumoniae*, *A. baumannii*, *E. aerogenes*, *Neisseria meningitidis*, and *C. albicans*. The remarkable antimicrobial activity of tannins is conditioned by their ability to bind to cell membranes through hydrophobic interactions via hydrogen bonding, which decreases membrane potential and increases permeability [72]. Due to the chelating of mineral macronutrients, such as iron or zinc, which are involved in bacterial metabolism and possess primarily electron transport function and catalytic or structural cofactors of many vital enzymes, tannins cause iron deprivation, thereby leading to irreversible cell death [73]. It is worth noting that the antimicrobial potential of tannins depends primarily on their chemical nature rather than on concentrations. Farha et al. [74] highlighted the superior antimicrobial activity of hydrolysable tannins, such as tannic acid, compared with monomeric catechin due to the abundance of the -OH groups in each of the ten galloyl groups. Given the results of the TTC content and the observations mentioned above, the prepared *V. opulus* water extracts can be regarded as a potential alternative for preventing bacterial infections.

The TAC values observed in this study are consistent with those reported by Kajszczak et al. [7] regarding the fruit of *V. opulus*, which were ranged from 6.0 to 53.0 mg CGE 100 mL$^{-1}$ FW. However, the TAC content in fresh *V. opulus* fruit was found to be three times higher, as reported by Česonienė et al. [23]. The fluctuations in the TAC content observed in the present study and those reported by other researchers can be explained by the relative susceptibility of these compounds to degradation under a pH higher than 7 and by differences found in the extraction solvents applied [75,76]. Interestingly, Demirdöven et al. [77] reported a similar TAC content observed in red cabbage extracts (*Brassica oleracea* L.), which was approached through optimized ultrasound-assisted extraction utilizing 42% ethanol as a sole solvent. It has been highlighted that some Gram-positive bacteria are more susceptible to anthocyanin action than Gram-negative bacteria [78]. The mechanisms underlying anthocyanin activity include the membrane and intracellular interactions of these compounds. The study by Lacombe et al. [79] demonstrated the ability of anthocyanin fractions derived from *Vaccinium macrocarpon* to contribute to the inhibition of *E. coli* O157:H7 at a native pH. In contrast, the antimicrobial activity was markedly reduced after a pH adjustment to neutral. The authors observed cell damage causing the leakage of nucleotides and other cytoplasm macromolecules, along with cell aggregation and disintegration, leading to death. Anthocyanins extracted from the fruit of *Punica granatum* L. and *Vaccinium vitis-idaea*, among others, contribute significantly to the inhibition of *E. coli* and *S. enterica*, as outlined in the review article of Ma et al. [80]. However, individual anthocyanin constituents that most exhibit antimicrobial activity cannot be identified since fruit extracts or juices are represented by a diversity of compounds such as organic acids, phenolic acids, flavonoids, anthocyanins, and saccharides that synergistically contribute to morphological changes in the bacterial cells, which leads to their irreversible death [78].

According to the obtained results, 18 individual, free phenolic compounds were qualitatively and quantitatively estimated after the purification of *V. opulus* extracts by taking advantage of the SPE technique and using the SPE "Supel™-Swift HLB" column. This study utilized the SPE as the high-molecular-weight compounds represented in plant matrices, such as proteins and polysaccharides, primarily affecting the correct quantification compounds of interest. The LC-ESI-TQ-MS/MS analysis of the *V. opulus* extracts revealed the presence of flavonols, such as quercetin, rutin, luteolin and its glucoside, kaempferol, rhamnetin, and isorhamnetin; flavan-3-ols such as catechin and epicatechin; and hydroxycinnamic and hydroxybenzoic acid derivatives, such as vanillin, gallic, neochlorogenic, cryptochlorogenic, chlorogenic, protocatechuic, caffeic, *trans*-ferulic, and *para*-coumaric acids. Although the identification of the anthocyanin group representatives in this study was impossible due to technical limitations, the distribution pattern of the phenolics in the *V. opulus* fruit was comparable to earlier studies by Kajszczak et al. [7], who indicated the presence of all four (plus anthocyanins) phenolic groups in the *V. opulus* fruit reported herein. Chlorogenic acid was found to be the most prevailing compound identified in the *V. opulus* extracts, and the most substantial contribution of the compound to the total amount of individual phenolics investigated was observed for the *V. opulus* extract derived from berries, which made up 90.3% of the total phenolics. Earlier, Velioglu et al. [3] also established that chlorogenic acid was a prevalent compound among 13 phenolics identified in the juice of *V. opulus*, accounting for 54% or 2037 mg kg$^{-1}$ of the total phenolics. This observation was further reinforced by Perova et al. [1], who indicated that its contribution to the total amount of phenolics could be up to 96.2%. Altun and Yilmaz [81] found an abundance of chlorogenic acid in the *V. opulus* fruit, followed by leaves, which supports the findings of the current study. The abundance of chlorogenic acid in the fruit and juice of *V. opulus* makes this object less attractive due to its remarkable bitterness and astringency [82,83]. However, this does not negate their use as active pharmaceutical ingredients [8]. Recent studies examining the antimicrobial activity of chlorogenic acid have revealed its outstanding antimicrobial and antibiofilm effects against *Yersinia enterocolitica* [84]. This observation was also reinforced by Lou et al. [85], who indicated its potential effectiveness in inhibiting three Gram-positive and three Gram-negative bacteria, including *S. pneumonia*, *S. aureus*, *B. subtills*, *E. coli*, *S. dysenteriae*, and *S. typhimurium*. It has been reported that chlorogenic acid interacts with the cell wall membrane, causing its permeabilization and partial leakage of nucleotides and other cytoplasm macromolecules, thus triggering cell inactivation [86]. However, it has been proposed that the loss of cell components only partially affects the viability of microorganisms. Some intracellular processes in bacteria, apart from cell wall permeabilization, lie behind the antimicrobial activity of chlorogenic acid [87]. The observations revealed the ability of chlorogenic acid to bind to shikimate pathway enzymes with high affinity and to inhibit their catalysis, thus causing irreversible cell death. In addition to chlorogenic acid, quinic acid derivatives, such as neochlorogenic and cryptochlorogenic acids, have also been found in the extracts of *V. opulus*. The highest contribution of neochlorogenic acid to the total individual phenolics was observed for the *V. opulus* extract derived from bark, which made up 11.1% of the phenolics investigated. Polka et al. [18] highlighted the superiority of bark over fruit and flowers by considering the distribution of neochlorogenic and cryptochlorogenic acids, thus reinforcing the present study's findings. Catechin, a phenolic compound, was found to be the second most abundant compound in the extracts of *V. opulus*. The highest catechin concentration was observed in the bark extract, accounting for 57.7% of the total phenolic content. These results are consistent with the findings of Zakłos-Szyda et al. [88], who reported that catechin is the main ingredient of *V. opulus* fresh juice and the phenolic-rich fraction, which corresponded to 121 mg 100 g$^{-1}$ FW and 100.764 mg 100 g$^{-1}$ DW, respectively. It is worth noting that the results of the present study contradict those reported by Polka et al. [18], who found no catechin in the *V. opulus* fruit and an excess in the flowers. The differences in the fruit maturity stage, flower physiology stage, and extraction techniques may have influenced the extractability of phenolics. As with chlorogenic acid,

monomeric catechin rather than galloylated forms possess promising antimicrobial activity against certain Gram-positive and Gram-negative bacteria [89]. The ability of catechins to partition into the lipid bilayers of various components through hydrophobic interactions via hydrogen bonding between the -OH groups of catechins and the O atoms of lipophiles leads to a lateral expansion of membranes, and cell permeability increases [89]. It has been experimentally established that antimicrobial activity, apart from reducing cell membrane potential, is conditioned by the formation of hydrogen peroxides through the oxidation of catechins, which leads to DNA damage and the oxidation of vital cell components such as organelles and membranes [90]. Given the observations made by other researchers, it is worthwhile to assume that the presence of catechin, along with epicatechin, in the *V. opulus* extracts that were obtained at high concentrations will substantially contribute to the inhibition of at least Gram-positive bacteria since the disability of negatively charged catechins to interact with model membranes of negative charge has been highlighted [91]. The presence of luteolin as the third most prevailing phenolic was observed in the extracts of *V. opulus*, and this is the first report revealing the existence of this type of flavonoid in *V. opulus* flowers; however, their presence has already been evidenced in *V. opulus* [92] fruit and leaves [61]. In a study on the antimicrobial activity of individual phenolics, the ability of flavonoids to inhibit the efflux pumps of *S. aureus* in a concentration-dependent manner has been established in the following order: myricetin > rhamnetin > kaempferol > apigenin > luteolin > quercetin [69,93]. Nevertheless, it should be stated that the other flavonoid representatives observed in the extracts of *V. opulus* were detected at relatively low quantities, and their contribution to the antimicrobial activity will likely be negligible.

Antioxidants have become scientifically fascinating substances due to their tremendous clinically proven benefits, which include diminishing oxidative stress, supporting disease prevention and eye health, aiding brain function, contributing to mental health improvement, reducing inflammation, and stimulating probiotic development within gut microbiomes [94,95]. The mechanisms of antioxidants (AOAs) are primarily but not limited to scavenging free radicals such as superoxide anions and hydroxyl radicals [96]. Given these statements, the ability of *V. opulus* extracts to quench free radicals as part of a further study was established by three commonly used AOA methods, i.e., DPPH$^\bullet$, FRAP, and ABTS$^{\bullet+}$. The AOA of the different morphological parts of the *V. opulus* extracts investigated was in the following order: berries > bark > mix > flowers. The results obtained by Polka and Podsędek [62] indicated that the AOA of *V. opulus* extracts is oriented toward hydroxyl radicals scavenging in the following effectiveness order: bark > flowers > fruits. Due to differences in expression units and extraction methods, the results of the present study are only in partial agreement with those reported by other researchers. Perova et al. [1] reported a broader variation in the AOA pinpointed by DPPH$^\bullet$, whereby they showed a range from 377.0 to 968.0 mg TE 100 g$^{-1}$ for dry *V. opulus* fruits. Barak et al. [97] observed a substantially higher AOA value for the water extract of *V. opulus* dried fruit, which corresponded to 96.7 mg of butylated hydroxytoluene equivalent (BHTE) g$^{-1}$. The obtained data showed a direct relationship between the AOA of *V. opulus* extracts and the TPC, TFC, and TTC values. However, a difference emerged between the AOA and individual phenolic compounds due to the distribution order and concentration of the individual phenolics observed in *V. opulus* morphological parts, which was as follows: berries > mix > bark > flowers. A similar AOA pattern was obtained by the FRAP method, which showed the contribution of the *V. opulus* obtained extracts to free radicals scavenging in the same order as was obtained by the DPPH$^\bullet$ method. This statement was additionally supported by an observation made in the earlier study by Muniyandi et al. [98], who reported that tannins are better radical scavengers than flavonoids and anthocyanins. Some studies have reported a strong correlation between DPPH$^\bullet$ and ABTS$^{\bullet+}$ and flavonoid values [68,99,100]. However, according to the results obtained on the content of the investigated individual phenolic compounds and those of DPPH$^\bullet$, FRAP, and ABTS$^{\bullet+}$, it can be assumed a direct role of catechin toward AOA rather than flavonoids. The most substantial reactivity of catechins against free radicals that are determined by FRAP and ABTS$^{\bullet+}$ was also reported

by Grzesik et al. [101], who revealed the superiority of monomeric catechin, epicatechin, and epigallocatechin gallates. Despite the extracts derived from *V. opulus* berries and a mix containing the highest chlorogenic acid, their contribution to AOA was negligible. No substantial contribution of either total individual saccharides or AAs was observed toward AOA.

The antimicrobial activity of the obtained *V. opulus* extracts against 19 test microorganisms was probed using the agar well diffusion method. The selection of this screening method to test antimicrobial activity was based on evidence regarding the lack of direct interconnection between the complex mixtures represented by high-molecular-weight compounds of a hydrophilic and lipophilic nature deposited on a disc with bacteria distributed over media. Due to the compounds' relatively low diffusivity magnitude, the Kirby–Bauer method only vaguely represents the antimicrobial activity. The initial screening showed that *V. opulus* extracts had no antimicrobial activity against certain microorganisms, including *L. monocytogenes*, *B. cereus*, *C. sakazakii*, *E. cloacae*, *C. perfringens*, *E. coli*, *C. freundii*, *C. albicans*, and *A. brasiliensis*. Sangma et al. [102] made a similar observation, thereby revealing less antimicrobial activity of the methanol extracts from the fruit and leaves of *V. simonsii* against Gram-negative *E. coli* and *S. enterica* and the microscopic fungus *C. ablicans*. The lack of antimicrobial activity of the *V. opulus* extracts that are rich in the compounds of negative surface charge can partially explain the resistance of Gram-negative bacteria, which is represented by a double membrane composed of phospholipids and lipopolysaccharides outside a thin peptidoglycan layer that is inside of a negative charge [91]. This statement reinforces the observation regarding the antimicrobial activity of the *V. opulus* extracts in inhibiting the Gram-negative bacteria selected in the current study. Fan et al. [103] indicated that catechins, for instance, have a stronger affinity to the peptidoglycan of Gram-positive bacteria than the negatively charged lipopolysaccharides in the membrane of Gram-negative bacteria. A remarkable activity of the *V. opulus* extract derived from berries against Gram-negative *C. muytjensii* and *P. aeruginosa* bacteria was revealed, which can be attributed to the abundance of the chlorogenic acid in this type of *V. opulus* extract and its ability to act on the intracellular membrane of *P. aeruginosa*, thereby causing a permeability increase, as well as detachment and loss of ATP [104]. A positive effect of catechin-rich *V. opulus* extracts derived from bark was also observed inhibiting *C. muytjensii*, thus supporting the lateral expansion of the membrane and increasing cell permeability. Among the *V. opulus* extracts investigated, the extract derived from berries exhibited the most remarkable antimicrobial activity against Gram-positive bacteria *S. aureus*. Düz et al. [67] reported a substantially lower activity of the ethanol and water extracts of *V. opulus* fruit against *S. aureus* despite the presence of a higher concentration of bioactive compounds than that was observed in the present study, the values of which corresponded to 18.6 and 14.0 mm. It is worth noting that the extracts obtained from *V. opulus,* except from flowers and berries, did not contribute to the suppression of *L. innocua* and *L. ivanovii* growth. These results are consistent with the findings of Puupponen-Pimiä et al. [105], who indicated the resistance of *L. monocytogenes* and *L. innocua* to all eight berry extracts. The lack of inhibition is due to *Listeria* spp. stress resistance and this genera's ability to survive under harsh environmental conditions such as high acidity; the presence of osmolytes, oxidants, and bacteriocins; low and high pressure; and UV light [106]. The presence of conjugative plasmids and transposons carrying antibiotic resistance makes up most of the *Listeria* spp. isolates from clinical, food-borne, and environmental sources resistant to topical antibiotics [107]. Overall, the extracts derived from *V. opulus* effectively inhibited the Gram-positive and Gram-negative bacteria selected in the present study. The produced inhibition zones were similar to those reported by Česonienė et al. [21] for *V. opulus* ethanol extracts, and it was several times higher than those reported by Adebayo et al. [108] for *V. opulus* water extracts.

The MIC values for the *V. opulus* extracts of different morphological parts were defined for the first time, generally reinforcing the data approached by the agar well diffusion method. This study indicates the remarkable inhibitory activity of the obtained *V. opulus*

extracts, wherein they acted in a homeopathic mode of action in a dose-dependent manner, thereby revealing that lower extract doses are more effective than higher ones. It is worth noting that the extract obtained from the flowers of *V. opulus* did not show any significant inhibitory activity against the selected bacteria, except for *L. innocua*, at either high or low extract doses. This suggests that *L. innocua* is highly susceptible to the constituents present in flowers rather than in the bark or berries of *V. opulus.* Additionally, the high concentration of AAs in the *V. opulus* extract derived from flowers may have encouraged bacterial growth rather than inhibiting it. At higher concentrations, the effects of obtained *V. opulus* extracts are less pronounced due to the presence of high-molecular-weight compounds such as pectic polysaccharides, proteins, and fats [18]. These compounds limit the access of *V. opulus* bioactives to the cell membranes, thus weakening the interaction between the bioactive substances and vital organs of the bacteria. This statement can be reinforced by an early observation made by Ildiz et al. [109], who indicated a lack of antimicrobial activity of free *V. opulus* fruit extracts at a concentration range of 2000–125 $\mu L\ mL^{-1}$ against four Gram-positive and four Gram-negative bacteria, which included two microscopic fungi. However, the same group of researchers reported that nano-scaled metallic objects engineered by green synthesis utilizing *V. opulus* fruit extracts deliver remarkable antimicrobial activity, which is achieved by the ability of additional negatively charged regions of nanoparticles to interact with the cell membranes more effectively, thus leading to cell membrane damage. Overall, the obtained agar well diffusion and MIC values indicate the homeopathic potential and effectiveness of *V. opulus* extracts in their original state in inhibiting Gram-positive and Gram-negative bacteria at relatively low concentrations. However, further studies are needed to pinpoint the molecular mechanisms of action.

## 5. Conclusions

In the present study, the first-time examination of the minimum inhibitory concentration (MIC) and inhibition zone values of industrially produced water extracts derived from various parts of the European cranberry bush *V. opulus* L., including its flowers, bark, berries, and a mixture thereof, was conducted. The selected extracts were tested against 19 cultures that underwent antibiotic susceptibility tests. The profiles of the saccharides, amino acids, and phenolics of the *V. opulus* extracts were pinpointed as a complement to the antimicrobial and antioxidant activity tests. The *V. opulus* extracts exhibited antimicrobial potential against 8 out of 19 test microorganisms and showed that the lowest effective inhibitory concentration values ranged from 0.24 and 0.49 $\mu L\ mL^{-1}$. The difference in the MIC values was conditioned by the nature of the extracts and the composition of bioactives. An example can be found in the extract derived from flowers, which specifically showed a high content of amino acids and proline, as well as a low content of phenolics that promoted microorganism proliferation instead of inhibition. In turn, the extracts derived from *V. opulus* bark, berries, and a mixture thereof, due to denser bioactive composition (which particularly apply to the number of flavan-3-ols and the hydroxycinnamate content), were effective against the selected bacteria. Moreover, the extracts exhibited a homeopathic mode of action, and the antimicrobial activity was dose-dependent. The antioxidant activity of the *V. opulus* extracts was tannin content-dependent, wherein it showed remarkable DPPH$^{\bullet}$, FRAP, and ABTS$^{\bullet+}$ radical scavenging potential. The *V. opulus* extracts obtained on a production scale have shown antimicrobial potency and an abundance of antioxidants, thus making it possible to regard them as a potential alternative for preventing bacterial infections.

**Supplementary Materials:** The following supporting information can be downloaded at: https://www.mdpi.com/article/10.3390/horticulturae10040367/s1, Table S1: Calibration data and linearity for group compounds and antioxidant activity tests; Table S2: Multiple reaction monitoring (MRM) transitions, collision energy, Q1, Q3 and dwell time for investigated amino acids; Table S3: Multiple reaction monitoring (MRM) transitions, collision energy, Q1, Q3 and dwell time for investigated phenolic compounds; Figure S1: Extracted ion chromatogram (EIC) in multiple reaction monitoring (MRM) represents the profile of 17 multiple amino acids identified in the extract derived from flowers

of *Viburnum opulus* L; Figure S2: Extracted ion chromatogram (EIC) in multiple reaction monitoring (MRM) represents the profile of 17 multiple amino acids identified in the extract derived from mixture of morphological parts (berries without seeds, leaves, buds and bark) of *Viburnum opulus* L; Figure S3: Extracted ion chromatogram (EIC) in multiple reaction monitoring (MRM) represents the profile of 17 multiple amino acids identified in the extract derived from bark of *Viburnum opulus* L; Figure S4: Extracted ion chromatogram (EIC) in multiple reaction monitoring (MRM) represents the profile of 17 multiple amino acids identified in the extract derived from berries of *Viburnum opulus* L; Figure S5: Extracted ion chromatogram (EIC) in multiple reaction monitoring (MRM) mode represents the profile of 18 phenolic standards at the concentration of 1 µg mL$^{-1}$; Figure S6: Extracted ion chromatogram (EIC) in multiple reaction monitoring (MRM) represents the profile of major phenolic compounds identified in the extract derived from flowers of *Viburnum opulus* L; Figure S7: Extracted ion chromatogram (EIC) in multiple reaction monitoring (MRM) represents the profile of major phenolic compounds identified in the extract derived from mixture of morphological parts (berries without seeds, leaves, buds and bark) of *Viburnum opulus* L; Figure S8: Extracted ion chromatogram (EIC) in multiple reaction monitoring (MRM) represents the profile of major phenolic compounds identified in the extracts derived from bark of *Viburnum opulus* L; Figure S9: Extracted ion chromatogram (EIC) in multiple reaction monitoring (MRM) represents the profile of major phenolic compounds identified in the extracts derived from berries of *Viburnum opulus* L.

**Author Contributions:** Conceptualization, K.J.-R. and V.R.; data curation, K.J.-R. and V.R.; formal analysis, K.J.-R., I.K. and V.R.; investigation, K.J.-R., I.K. and V.R.; methodology, K.J-R., I.K. and V.R.; resources, D.S., A.V. and V.R.; software, V.R.; visualization, K.J.-R. and V.R.; writing—original draft, V.R.; writing—review and editing, K.J.-R., D.S., A.V., S.M.-B. and V.R. All authors have read and agreed to the published version of the manuscript.

**Funding:** This research received no external funding.

**Data Availability Statement:** Data are contained within the article and Supplementary Materials.

**Conflicts of Interest:** The authors declare no conflicts of interest.

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
