# Peer review of "Scrutinizing the Antimicrobial and Antioxidant Potency of European Cranberry Bush (Viburnum opulus L.) Extracts"

_horticulturae, doi:10.3390/horticulturae10040367_

Round 1

Reviewer 1 Report

Comments and Suggestions for Authors

Although the presented information is interesting, several modifications must be done to be considered for publication in this prestigious Journal

Title of the Subsection 2.3 is too long; it should be shorter. Same case in Subsection 2.9.

Folin-Ciocalteu method is not a specific method for the determination of polyphenolic compounds, as this reagent can react with other compounds such as vitamin C. How was overestimation avoided?

Antioxidant activity should not be judged by a single method or methods involving similar mechanism (as DPPH and ABTS assays) due to the multifactorial reactions involved in these reactions. Then, AA should be determined by other methods (complementary to FRAP and the free radical scavenging assays). Otherwise, free radical scavenging and reducing power terms should be used.

In the sentence: “Presumably, prolonged ultrasonic treatment promoted the release of structurally bound cell components, including xylose moieties, as reported by Wang et al. [41].” Why? Explan.

In the sentence: “The presence of alanine in the extracts of VO, flower extract, specifically, makes it attainable and worthwhile to assume the possible contribution of this extract to the growth of selected opportunistic bacteria as the extensive proliferation of E. coli K-12 and EC-14 has been observed by Muranaka et al. [53] in infected model mice with an excess of alanine”. Explain.

Subsections 3.3, 3.4, 3.5, 3.6 and 3.7 are well discussed. This could be applied in other subsections of the Results and discussion.

Comments on the Quality of English Language

Manuscript is difficult to follow due to English language. In this sense, it should be improved.

Author Response

Response to Reviewer's 1 comments

R1: Although the presented information is interesting, several modifications must be done to be considered for publication in this prestigious Journal

A: The authors thank the Reviewer for carefully checking our manuscript and for valuable comments. The authors have incorporated most of the suggested changes in the manuscript. The authors refer to them in detail below.

R2: Title of the Subsection 2.3 is too long; it should be shorter. Same case in Subsection 2.9.

A: The authors would like to thank the Reviewer for valuable remark. We notify you that the Sections and Subsections of the manuscript were revised accordingly. 

R3: Folin-Ciocalteu method is not a specific method for the determination of polyphenolic compounds, as this reagent can react with other compounds such as vitamin C. How was overestimation avoided?

A: The authors fully agree with the Reviewer's point. Therefore, to verify the results on bioactive compounds, the present study utilized selective analysis of purified SPE extracts approached by LC-ESI-TQ-MS/MS. The data obtained both by spectrophotometry and mass-spectrometry were compared and correlated. Please refer to Table 6. 

R4: Antioxidant activity should not be judged by a single method or methods involving similar mechanism (as DPPH and ABTS assays) due to the multifactorial reactions involved in these reactions. Then, AA should be determined by other methods (complementary to FRAP and the free radical scavenging assays). Otherwise, free radical scavenging and reducing power terms should be used.

A: The authors fully agree with the Reviewer's point. Therefore, the assessment of AA has been approached by three independent methods: 2.6.1. DPPH• Free Radical Scavenging Activity, 2.6.2. Ferric Reducing Antioxidant Power (FRAP), and 2.6.3. ABTS•+ Radical Cation Scavenging Activity. To make the results easier for readers to understand, the authors use the general term antioxidant activity (AOA) of extracts. 

R5: In the sentence: "Presumably, prolonged ultrasonic treatment promoted the release of structurally bound cell components, including xylose moieties, as reported by Wang et al. [41]." Why? Explan. 

A: The release of bound components is conditioned by ultrasound's ability to increase cell wall permeability and eventually disrupt it, leading to a more efficient mass transfer between the solvent and plant material. This can be reinforced by an early observation made by Sun et al. [43]. https://doi.org/10.1016/j.fbio.2022.101646.

R6: In the sentence: "The presence of alanine in the extracts of VO, flower extract, specifically, makes it attainable and worthwhile to assume the possible contribution of this extract to the growth of selected opportunistic bacteria as the extensive proliferation of E. coli K-12 and EC-14 has been observed by Muranaka et al. [53] in infected model mice with an excess of alanine". Explain.

A: Dear reviewer! The authors assume that amino acids, particularly alanine, that are abundantly observed in the V. opulus extract derived from flowers directly or indirectly contribute to the proliferation of microorganisms and not their inhibition. The authors' assumption is reinforced by findings made by:

1 Díaz-Pascual F, Lempp M, Nosho K, Jeckel H, Jo JK, Neuhaus K, Hartmann R, Jelli E, Hansen MF, Price-Whelan A, Dietrich LE, Link H, and Drescher K. Spatial alanine metabolism determines local growth dynamics of escherichia coli colonies. Elife 10:1–29 (2021).

2 Wei Y, Qiu W, Zhou XD, Zheng X, Zhang KK, Wang S Da, Li YQ, Cheng L, Li JY, Xu X, and Li MY. Alanine racemase is essential for the growth and interspecies competitiveness of Streptococcus mutans. Int J Oral Sci Nature Publishing Group; 8:231–238 (2016).

R7: Subsections 3.3, 3.4, 3.5, 3.6 and 3.7 are well discussed. This could be applied in other subsections of the Results and discussion.

A: The authors tried to adhere to the same writing style when preparing the manuscript.

R8: Manuscript is difficult to follow due to English language. In this sense, it should be improved.

A: The authors understand the Reviewer's concern. The authors have double-checked the manuscript's spelling to make the content easier to read. 

Reviewer 2 Report

Comments and Suggestions for Authors

This work reports the antimicrobial and antioxidant activities of Cranberry Bush (Viburnum opulus L.) water extracts of different parts. The chemical composition was also qualitatively and quantitatively analyzed. The four studied extracts exhibited antimicrobial and antioxidant activities to a different extents, depending on their chemical composition. 

General comments: 

- In the results and discussion section, the authors started with the chemical composition that they used to discuss the antimicrobial composition. However, since the results of that activity were not reported yet, it was difficult to follow. To make it clearer to readers, authors can either put these ideas in the section reporting the antimicrobial activity results, or separate results and discussion into two sections.

- Please avoid adding abbreviations that are not used several times in the manuscript; antimicrobial resistance (AMR), active pharmaceutical ingredients (APIs).

Specific comments:

Abstract

- The Abstract should begin with the goal of the work, then the methods used and the results.

- "The examination of Viburnum opulus L. (VO) water extracts derived from flowers, a mixture of morphological parts, i.e., flowers, bark, berries, and their mixture (berries without seeds, leaves, buds and bark)":  the parts mentioned here are different than the ones in material and methods, please correct.

- Please add the technique used to quantify phenolics, flavonoids, tannins, and anthocyanins.

- "Profiling of individual phenolic compounds disclosed the superiority of chlorogenic acid (up to 90.3%) in berry and mix extracts, catechin (up to 57.7%) and neochlorogenic acid (11.1%) in bark extract, which conveyed a remarkable contribution towards antimicrobial activity": The discussion of the antimicrobial activity should come after reporting the results of that activity.

- "Owing to substantially denser bioactive composition, the VO berries and bark extracts exhibited markedly better AOA": using "better" means that the activity is compared to something.

- "VO extracts derived from mix, bark, and berries were more active against 8 out of 19 selected test microorganisms at concentrations from 0.24 to 0.49 µL mL−1 than at 500.0 µL mL−1": These concentrations are not clear, please explain (also in the results and discussion section)

- It may be interesting to add the saccharide profile.

- Keywords: consider adding the antioxidant activity.

Introduction:

- Several parts seem more like discussion, please revise.

- "Moreover, the antimicrobial activity of VO juice was also emphasized by conducting an analysis of antioxidant activity against Salmonella Agona, B. subtilis, Listeria monocytogenes, Enterococcus faecalis, Micrococcus luteus, S. epidermidis [20]": what do you mean by antioxidant activity?

Material and Methods

- 2.2. 

Please clarify if the plant was bought as raw material or an extract prepared somehow (the information on the extraction preparation steps is unavailable : that's problematic).

Please explain how the extract was prepared for antimicrobial activity.

- 2.4: 

"Briefly, 3 mL of VO berry (4) or 5 mL of flowers (1), mix (2), or bark (4) extract ..." Please explain why different volumes were used. Please correct bark extract (3 not 4).

Please make it clear that the saccharide profile was analyzed too.

- 2.11.1:

Please specify if the well diameter is included in the results or not.

Results and Discussion

- 3.1. 

Please consider another opening idea for this section.

"According to this observation, the first step in analyzing VO extracts was establishing the sugar profile and content": it's not clear from material and methods that it was the first step.

"Like other saccharides, the highest value was observed in berry-derived extract, while the lowest was in the mixture of VO parts": is this paragraph still talking about ribose? if yes, please merge it with the idea before.

" This observation reinforces the funding of the present study since the extract was prepared using all morphological parts of the VO shrub, i.e., flowers, whole berries, shoots and leaves, buds and bark": in the material and methods section, the parts mentioned in mix extract (2) were: flowers, berries without seeds, leaves, buds and bark. Please rectify.

- 3.2.

Table 3: Please specify in the table which Amino Acids are essential and which are branched. Is there a reason why the sums were calculated separately?

"The second prevalent AA identified in the extracts of the VO was glutamic acid, contributing to the total amount of AAs from 5.2 to 29.2% or from 50.2 to 156.6 mg 100 mL−1 in the VO extract derived from flowers and berries, respectively" This AA was dominant in bark not flowers.

"A recent study revealed disparities in AA metabolism in C. albicans biofilms, with a prominent upregulation of arginine, proline, aspartate, and glutamate metabolism in high biofilm-forming isolates [52]." Please explain why this information is relevant to your findings.

Figure 3: This figure repeats the results of Table 3. It is better to put it in supplementary data.

"Overall, from a nutritional standpoint, the extracts derived from the flowers and mix of VO can be highlighted as exceptional as AA levels are many times higher than that reported by other researchers for most common fruit juices." The extracts and juices will have different compositions because they're different, please compare with extracts, not juices.

- 3.3.

"Regarding the outstanding TPC content in VO extracts derived from berries and bark, their contribution towards the inhibition of opportunistic microorganisms would be expected, as was already reported by Mahboubi et al. [64] and Ispiryan et al. [65] for Punica granatum L. and Rubus idaeus L. extracts, respectively." Please discuss your results with the same species or genus.

- 3.7.

"Quinic acid derivatives such as neochlorogenic and cryptochlorogenic acids have also been observed in the extracts of VO": Those components are in low concentration, please explain why they're cited in the second position in the reading.

"The contribution of neochlorogenic to the total phenolics was 11.1%." Please specify the extract mentioned here.

"Polka et al. [17] highlighted the superiority of bark over fruit and flowers" the superiority in what? please specify.

"The presence of luteolin as the third most prevailing phenolic was observed in the extracts of VO in the range from 0.01 to 1.15 mg 100 mL−1 FW": please specify that it's the flower extract.

"Nevertheless, it should be stated that the other flavonoid representatives observed in the extracts of VO were detected at relatively low quantities, and their contribution to the antimicrobial activity will likely be negligible." Even though the concentration is low, other flavonoids can contribute (synergy or addition) to the activity.

- 3.8

Figure 4: Please verify the statistic: DPPH results (see also FRAP) of berries and bark extracts seem similar and thus should have the same letter.

- 3.9. 

"Further analysis revealed remarkable activity of VO extract derived from VO berries against Gram-negative C. muytjensii and Gram-positive P. aeruginosa bacteria, producing zones of inhibition 28.6 and 22.4 mm, respectively (Table 7)." P. aeruginosa is a Gram-negative bacteria.

Table 7: please add the Gram type in the table (also Table 8). Also, specify if the well diameter was included in the results. Please correct:" *The MIC of amoxicillin/ clavulanic acid at concentration 500.0 mg mL−1 was used as a positive control in all experiments." 

"This effect may be attributed to the availability of chlorogenic acid in the extract of VO derived from berries and its ability to act on the intracellular membrane of P. aeruginosa, causing its permeability increase and detachment and loss of ATP [104]. " please link this paragraph with the previous one since it finishes the idea.

"The produced zones of inhibition are several times higher than that reported by Balčiūnaitienė et al. [108] for ethanolic extracts of medicinal plants Artemisia absinthium L., Humulus lupulus L. and Thymus vulgaris L. and by Yildirim et al. [109] for essential oils of Achillea millefolium L. and Achillea wilhelmsii L. plants." Please compare the results with works on the same species or genus.

- 3.10.

Figure 5. This figure repeats the results presented in the table and is not clear enough, please eliminate it.

"An earlier study by Shai et al. [111] examined the antimicrobial activity of 16 different plant extracts native to southern Africa and observed superior activity of acetone extracts derived from medicinal plants of Xanthorcesis zambesiaca and Cassia abbreviate, showing effective MIC values of 0.113 and 0.285 mg mL−1, respectively" please explain the reason why you mention this work. Similarly, it reports the results of other plants very far away from the selected one here.

Remark: Please consider adding the line numbers to make it easier to refer to the parts in the manuscript.

Comments on the Quality of English Language

The quality of the English Language was generally good. Some minor corrections are needed.

- Abstract: "While the lowest content of individual phenolics was in flower extract." 

- Results and discussion : 

3.1.

"Mizzi et al. [38] conveyed as the lowest required to inhibit the growth of S. aureus and E. coli": lowest concentration

"Cleaver et al. [49] indicating the of role respiratory substrate"

3.5.

"The concertation of TTC content": please correct: concentration

3.7.

"while 12.85 mg 100 mL−1 FW in berry extract": please finish the sentence

Author Response

Response to Reviewer’s 2 comments

R1: This work reports the antimicrobial and antioxidant activities of Cranberry Bush (Viburnum opulus L.) water extracts of different parts. The chemical composition was also qualitatively and quantitatively analyzed. The four studied extracts exhibited antimicrobial and antioxidant activities to a different extents, depending on their chemical composition. 

A: The authors thank the Reviewer for carefully checking our manuscript and for valuable comments. The authors have incorporated most of the suggested changes in the manuscript. The authors refer to them in detail below.

General comments: 

In the results and discussion section, the authors started with the chemical composition that they used to discuss the antimicrobial composition. However, since the results of that activity were not reported yet, it was difficult to follow. To make it clearer to readers, authors can either put these ideas in the section reporting the antimicrobial activity results, or separate results and discussion into two sections.

A: The authors appreciate the Reviewer's remark. However, they disagree with the suggestion of reporting the antimicrobial activity results first. They believe that a more reasonable and logical way to discuss the chemical composition of V. opulus is to first and only then interconnect these results with antimicrobial activity data, not vice versa. That is what was done in the revised version of the manuscript.
The authors hope that the Reviewer find this argument satisfactory. 

R2: - Please avoid adding abbreviations that are not used several times in the manuscript; antimicrobial resistance (AMR), active pharmaceutical ingredients (APIs).

A: The authors thank the Reviewer for their valuable remark. Abbreviations that were not used in the manuscript several times were removed. 

R3: Specific comments:

R3: Abstract

- The Abstract should begin with the goal of the work, then the methods used and the results.

A: The Abstract now contains the goal of the work, which turns into observations.

R4: - “The examination of Viburnum opulus L. (VO) water extracts derived from flowers, a mixture of morphological parts, i.e., flowers, bark, berries, and their mixture (berries without seeds, leaves, buds and bark)”: the parts mentioned here are different than the ones in material and methods, please correct.

A: This fragment has been revised to make it easier to understand the main object of the work.

R5: - Please add the technique used to quantify phenolics, flavonoids, tannins, and anthocyanins.

A: Dear Reviewer. The mentioned techniques were initially described in the original version of the manuscript. Please refer to 2.5. Spectrophotometric Studies section, 2.5.1. – 2.5.4. subsections 

R6: - “Profiling of individual phenolic compounds disclosed the superiority of chlorogenic acid (up to 90.3%) in berry and mix extracts, catechin (up to 57.7%) and neochlorogenic acid (11.1%) in bark extract, which conveyed a remarkable contribution towards antimicrobial activity”: The discussion of the antimicrobial activity should come after reporting the results of that activity.

A: The authors agree with the Reviewer’s point. The results of antimicrobial activity were introduced immediately after the chemical composition results.

R7:  - “Owing to substantially denser bioactive composition, the VO berries and bark extracts exhibited markedly better AOA”: using “better” means that the activity is compared to something.

 A: The authors understand the Reviewer’s concern. We have added clarification to this statement. Please refer to Lines 26 – 28 of the Abstract section. In the expression “better AOA” the authors intended to point out the superiority of Vopulus berries and bark extracts over flowers or a mixture of Vopulus morphological parts.  

R8: - “VO extracts derived from mix, bark, and berries were more active against 8 out of 19 selected test microorganisms at concentrations from 0.24 to 0.49 µL mL−1 than at 500.0 µL mL−1”: These concentrations are not clear, please explain (also in the results and discussion section).

A:  Dear Reviewer! The concentrations mentioned are the amount of extract in µL achieved by serial dilution of the stock (0.15 mL in the first well). Since the extract introduced to the first well was mixed with other ingredients to reach 0.3 mL volume, the real extract concentration in the first well is 0.5 mL mL−1, which in the step-wise dilution order (the same dilution factor) reduced to 0.24 mL−1. The sample preparation steps are described in detail: “2.11.2. Minimum Inhibitory Concentration (MIC)”. 

R9: - It may be interesting to add the saccharide profile.

A: Dear reviewer! The authors are happy that the Reviewer found this information interesting, but it could not be introduced within the Abstract section due to space limitations.  

R10: Keywords: consider adding the antioxidant activity.

A: The “antioxidant activity” is already present in the title of the manuscript.

R11: - Several parts seem more like discussion, please revise.

- “Moreover, the antimicrobial activity of VO juice was also emphasized by conducting an analysis of antioxidant activity against Salmonella Agona, B. subtilis, Listeria monocytogenes, Enterococcus faecalis, Micrococcus luteus, S. epidermidis [20]”: what do you mean by antioxidant activity?

A: Dear reviewer! The authors are grateful for the valuable observation and very sorry for confusing the Reviewer with the “antioxidant activity” term. The authors intended to point antimicrobial potential instead of antioxidant.

R12: Material and Methods

- 2.2. Please clarify if the plant was bought as raw material or an extract prepared somehow (the information on the extraction preparation steps is unavailable : that’s problematic).

A: Dear Reviewer! The object of the present study is “industrially produced V. opulus extracts rather than raw material”. This explains the lack of the “extraction procedure”. Please refer to 2.2. Sample Information section to find the information about the samples

R13: Please explain how the extract was prepared for antimicrobial activity.

A: Dear reviewer! No specific preparation other than filtering under aseptic conditions was done not to affect the microbiological status of the extract.

R14: - 2.4: “Briefly, 3 mL of VO berry (4) or 5 mL of flowers (1), mix (2), or bark (4) extract ...” Please explain why different volumes were used. 

A: Dear Reviewer! The difference in the volumes of extracts used for sample preparation is conditioned by their chemical composition and abundance of constituents. To avoid overloading equipment and stay within the calibration curves’ frame, berry extract needed to dissolve with more volume of solvent.  

R15: Please correct bark extract (3 not 4).

A: The authors are grateful to the Reviewer’s observation. This shortcoming has been corrected.

R16: Please make it clear that the saccharide profile was analyzed too.

A: The authors apologize for omitting this information. Now, this information is presented within Lines 133-137. 

R17: - 2.11.1: Please specify if the well diameter is included in the results or not.

A: This information is now indicated in the Materials and Methods section within Lines: 298-299.

Results and Discussion- 3.1.

R18: Please consider another opening idea for this section. “According to this observation, the first step in analyzing VO extracts was establishing the sugar profile and content”: it’s not clear from material and methods that it was the first step.

A: The authors highly appreciate the Reviewer’s point. We notify that the Materials and Methods section has been revised to be consistent with the Results and Discussion section. 

R19: “Like other saccharides, the highest value was observed in berry-derived extract, while the lowest was in the mixture of VO parts”: is this paragraph still talking about ribose? if yes, please merge it with the idea before.

A: The authors are grateful for the Reviewer’s remark. The authors are still talking about the ribose. This fragment has been merged with the previous one. 

R20: “This observation reinforces the funding of the present study since the extract was prepared using all morphological parts of the VO shrub, i.e., flowers, whole berries, shoots and leaves, buds and bark”: in the material and methods section, the parts mentioned in mix extract (2) were: flowers, berries without seeds, leaves, buds and bark. Please rectify.

A: Corrected. Thank you!

- 3.2.

R21: Table 3: Please specify in the table which Amino Acids are essential and which are branched. Is there a reason why the sums were calculated separately?

A: Additional information regarding groups of amino acids has been ensured. There is no specific reason to show the sums of amino acid groups, as this is only to emphasize the difference between the sample groups. 

R22: “The second prevalent AA identified in the extracts of the VO was glutamic acid, contributing to the total amount of AAs from 5.2 to 29.2% or from 50.2 to 156.6 mg 100 mL−1 in the VO extract derived from flowers and berries, respectively” This AA was dominant in bark not flowers.

A: Apparently, the Reviewer misunderstood the discussion of results since the authors do not state that the highest AA was found in flowers. This sentence intended to show the range of AA observed in four groups, which is from 5.2 to 29.2% or from 50.2 to 156.6 mg 100 mL−1 in the VO extract derived from flowers and berries, respectively”. We are using “respectively” to indicate the order of AA content in these samples. The highest concentration of glutamic acid was found in berry extract, contributing to the total AAs, which amount to 29.2%.

R23: “A recent study revealed disparities in AA metabolism in C. albicans biofilms, with a prominent upregulation of arginine, proline, aspartate, and glutamate metabolism in high biofilm-forming isolates [52].” Please explain why this information is relevant to your findings.

A: The authors intended to highlight the importance of amino acids in the metabolism of microorganisms with biofilm-forming ability.  

R24: Figure 3: This figure repeats the results of Table 3. It is better to put it in supplementary data.

A: The authors are grateful for the Reviewer’s remark. This figure was moved to the Supplementary Materials.

R25: “Overall, from a nutritional standpoint, the extracts derived from the flowers and mix of VO can be highlighted as exceptional as AA levels are many times higher than that reported by other researchers for most common fruit juices.” The extracts and juices will have different compositions because they’re different, please compare with extracts, not juices.

A: The authors understand the Reviewer’s concern. Unfortunately, no direct comparison of AA content can be made due to the limitations of the scientific literature. This is the first study that analysed the content of AAs in different morphological parts of V. opulus.  

R26: - 3.3. “Regarding the outstanding TPC content in VO extracts derived from berries and bark, their contribution towards the inhibition of opportunistic microorganisms would be expected, as was already reported by Mahboubi et al. [64] and Ispiryan et al. [65] for Punica granatum L. and Rubus idaeus L. extracts, respectively.” Please discuss your results with the same species or genus.

A: Dear Reviewer! This comparison is intended to prepare the reader for the results of antimicrobial activity. This is done in further discussion.

R27: - 3.7.” Quinic acid derivatives such as neochlorogenic and cryptochlorogenic acids have also been observed in the extracts of VO”: Those components are in low concentration, please explain why they’re cited in the second position in the reading.

A: Dear Reviewer. Those compounds are functionally related to chlorogenic acid, which was found to be the main compound observed in all four V. opolus extracts. The authors found it more logical to mention it after chlorogenic acid than at the end of this section. 

R28: “The contribution of neochlorogenic to the total phenolics was 11.1%.” Please specify the extract mentioned here.

A: The authors are grateful for the Reviewer’s remark. We have specified the extract type that contained the highest amount of neochlorogenic acid. Lines 619-621.

R29: “Polka et al. [17] highlighted the superiority of bark over fruit and flowers” the superiority in what? please specify.

A: Dear Reviewer! The authors already indicate that Polka et al. [18] highlighted the superiority of bark over fruit and flowers, considering the distribution of neochlorogenic and cryptochlorogenic acids, thus reinforcing the present study’s findings. 

R30: “The presence of luteolin as the third most prevailing phenolic was observed in the extracts of VO in the range from 0.01 to 1.15 mg 100 mL−1 FW”: please specify that it’s the flower extract.

A: This information is already highlighted in the following fragment: “The presence of luteolin as the third most prevailing phenolic was observed in the extracts of V. opulus in the range from 0.01 to 1.15 mg 100 mL−1 FW, with extract derived from flowers having the highest amount and extract derived from berries and bark the lowest.”

R31: “Nevertheless, it should be stated that the other flavonoid representatives observed in the extracts of VO were detected at relatively low quantities, and their contribution to the antimicrobial activity will likely be negligible.” Even though the concentration is low, other flavonoids can contribute (synergy or addition) to the activity.

A: The authors acknowledge the Reviewer’s remark. Currently, the authors have data that allows them to speculate about the influence of individual compounds on the growth of microorganisms rather than making a definite claim. Therefore, in their conclusion, the authors state that the influence of these compounds “will likely be negligible.” It’s important to note that this study did not involve testing individual isolates from Viburnum opulus L. for their antimicrobial activity. However, based on the data from the correlation (please refer to Table 8) between antioxidant activity (AOA) values and concentration of group compounds, tannins are directly contributing to AOA. This is reinforced by the correlation of individual compounds with AOA values, with catechin (CT) being particularly noteworthy.

R32: - 3.8 Figure 4: Please verify the statistic: DPPH results (see also FRAP) of berries and bark extracts seem similar and thus should have the same letter.

A: Dear Reviewer. The statistics are correct. The AOA by DPPH and FRAP for berry and bark extracts corresponds to 516.5 and 476.0 and to 249.3 and 229.5 TE mg 100 mL-1, respectively.

R33: - 3.9. “Further analysis revealed remarkable activity of VO extract derived from VO berries against Gram-negative C. muytjensii and Gram-positive P. aeruginosa bacteria, producing zones of inhibition 28.6 and 22.4 mm, respectively (Table 7).” P. aeruginosa is a Gram-negative bacteria.

A: The authors have corrected this shortcoming. Thank you!

R34: Table 7: please add the Gram type in the table (also Table 8). Also, specify if the well diameter was included in the results. 

A: The authors highly appreciate the Reviewer’s suggestion. We have added the Gram type to the Table 7 and Table 8

R35: Please correct:” *The MIC of amoxicillin/ clavulanic acid at concentration 500.0 mg mL−1 was used as a positive control in all experiments.”

A: The authors highly appreciate the Reviewer’s suggestion. We have revised the following sentence, and now it appears as follows: “*The MIC of amoxicillin/ clavulanic acid at a concentration range of 4.37 to 0.002 and 0.007 to 0.00003 mg mL−1 was used as a positive control, respectively.”

R36: “This effect may be attributed to the availability of chlorogenic acid in the extract of VO derived from berries and its ability to act on the intracellular membrane of P. aeruginosa, causing its permeability increase and detachment and loss of ATP [104].” please link this paragraph with the previous one since it finishes the idea.

A: Corrected. Thank you!

R37: “The produced zones of inhibition are several times higher than that reported by Balčiūnaitienė et al. [108] for ethanolic extracts of medicinal plants Artemisia absinthium L., Humulus lupulus L. and Thymus vulgaris L. and by Yildirim et al. [109] for essential oils of Achillea millefolium L. and Achillea wilhelmsii L. plants.” Please compare the results with works on the same species or genus.

A: Dear Reviewer! The authors would be grateful to retain this fragment without the change since it emphasizes the superiority of investigated extracts over the other medicinal plants.

R38: R39: - 3.10. Figure 5. This figure repeats the results presented in the table and is not clear enough, please eliminate it.

A: Dear reviewer! We have incorporated additional explanation to make this figure easier to understand. 

R39: “An earlier study by Shai et al. [111] examined the antimicrobial activity of 16 different plant extracts native to southern Africa and observed superior activity of acetone extracts derived from medicinal plants of Xanthorcesis zambesiaca and Cassia abbreviate, showing effective MIC values of 0.113 and 0.285 mg mL−1, respectively” please explain the reason why you mention this work. Similarly, it reports the results of other plants very far away from the selected one here.

A: We agree with the Reviewer’s point. However, it is impossible to compare the results obtained in this article with the literature due to the very limited information available to date covering the values of MIC for V. opulus extracts. We would be grateful to retain this comparison without the change. Thank you!

R40: Remark: Please consider adding the line numbers to make it easier to refer to the parts in the manuscript.

A: Dear Reviewer. We have added continuous line numbering in the manuscript. Thank you!

D41: - Results and discussion: 

3.1. “Mizzi et al. [38] conveyed as the lowest required to inhibit the growth of S. aureus and E. coli”: lowest concentration

A: Dear Reviewer. This fragment was revised. Thank you!

R42: “Cleaver et al. [49] indicating the role of respiratory substrate”

A: Dear Reviewer. This fragment was revised. Thank you!

R43: 3.5. “The concertation of TTC content”: please correct: concentration

A: Dear Reviewer. This fragment was revised. Thank you!

R44: 3.7. “while 12.85 mg 100 mL−1 FW in berry extract”: please finish the sentence.

A: Dear Reviewer. This sentence is finished: “The catechin concentration in this sample amounted to 24.82 mg 100 mL−1 FW, while 12.85 mg 100 mL−1 FW in berry extract.”

Reviewer 3 Report

Comments and Suggestions for Authors

Please, refer to PDF file.

Author Response

Response to Reviewer's 3 comments

R1: The manuscript is well-founded and covers aspects of research surrounding of V. opulus. The objectives are well aligned with previous studies and complement the efforts regarding the plant. The methodology is appropriate and very well defined. The results are well described and the conclusion adequately summarizes the results obtained and applicability of commercial extracts of V. opulus.

A: The authors thank the Reviewer for carefully checking our manuscript and for valuable comments. The authors have incorporated most of the suggested changes in the manuscript. The authors refer to them in detail below.

R2: As a suggestion, the authors could add the concentration ranges of each standard stock solution and the regression curves for each spectrophotometric procedure (sections 2.5 and 2.6).

A: Dear reviewer! Additional information with calibration data and linearity for group compounds (2.5.1. – 2.5.4.) and antiradical activity tests (2.6.1. – 2.6.3.) is now indicated in Table S2 as Supplementary Material.

R3: Prior to the ANOVA, normality and especially homoscedasticity were assessed using which methods? Did any variable require any transformation? This information is important because some values in the tables suggest a strong departure from homoscedasticity, such as:

  1. Table 2: Ribose (Bark and Berries), Fructose (Mix and Bark);
  2. Table 3: Aspartic acid (Flowers and Berries), Glutamic acid (Mix and Bark);
  3. Table 5: Chlorogenic acid (Mix and Bark).

A: Dear Reviewer! The authors would like to inform you that we analysed all the data obtained during the system’s calibration by performing a linear least squares regression of the instrument response versus the concentration of the respective standard, using integrated LabSolutions Insight” LC-MS software version 3.7 SP3 (workstation). We found that the calibration curves are linear, and R2 values were over .99, indicating a strong correlation, and RSD below 12%, indicating good repeatability and homoscedasticity. However, we observed heteroscedasticity in our model for variables obtained by the instrument after injecting samples due to the nature of the samples, which is generally a biological variance. We did not make any data transformations. The difference between chemical components and compounds of tested sample groups (4) allows us to distinguish specific compounds’ superiority that can be used as biomarkers during the research of one or another morphological part of Viburnum opulus L. Our research aimed to emphasize the difference between the biological value of the tested industrially produced Viburnum opulus L. products.

R4: The following text on page 18 (Nevertheless, it should be stated that the other flavonoid representatives observed in the extracts of VO were detected at relatively low quantities, and their contribution to the antimicrobial activity will likely be negligible.), suggests the correlation between concentration of metabolites and (biological) activity in the extracts. However, the text is closer to speculation than an experimentally based argument and should be revised. Although the presented information is interesting, several modifications must be done to be considered for publication in this prestigious Journal.

A: The authors acknowledge the Reviewer’s remark. Currently, the authors have data that allows them to speculate about the influence of individual compounds on the growth of microorganisms rather than making a definite claim. Therefore, in their conclusion, the authors state that the influence of these compounds “will likely be negligible.” It is important to note that this study did not involve testing individual isolates from Viburnum opulus L. for their antimicrobial activity. However, based on the data from the correlation (please refer to Table 8) between antioxidant activity (AOA) values and concentration of group compounds, tannins are directly contributing to AOA. This is reinforced by the correlation of individual compounds with AOA values, with catechin (CT) being particularly noteworthy.

Reviewer 4 Report

Comments and Suggestions for Authors

Author Response

Response to Reviewer's 4 comments

A: The authors thank the Reviewer for carefully checking our manuscript and for valuable comments. The authors have incorporated most of the suggested changes in the manuscript. The authors refer to them in detail below.

R1: Replace this phrase with "The widest range of all four group compounds was detected in the VO extracts sourced from berries, whereas the narrowest range was found in those obtained from flowers."

A: The authors are grateful to the Reviewer for the suggestion. We confirm replacing this phrase with the proposed one.

R2: Add some other references in this regards

A: Additional references were ensured. Thank you!

R3: Repalce with "Furthermore, Dursun et al. [16] emphasized the significant impact of extraction methods—such as ultrasound-assisted, microwave-assisted, Soxhlet, and solvent-based—on both the yield and quality of VO extracts."

A: The authors are grateful to the Reviewer for the suggestion. We confirm replacement of this sentence with proposed.

R4: Replace with "Thus, this study's primary goal was to examine four commercially available Viburnum opulus L. extracts that were obtained from (1) flowers; (2) a combination of morphological parts (flowers, berries without seeds, leaves, buds, and bark); (3) bark; and (4) berries without seeds. To achieve this, a qualitative and quantitative analysis of the bioactives profile and antimicrobial susceptibility test (MIC analysis) of reference test cultures were conducted, and the lowest concentrations that could inhibit the growth of the samples were determined."

A: The authors appreciate the Reviewer's suggestion. We confirm the inclusion of this sentence in the Introduction sections as suggested.

A1: The brake page was removed. Thank you!

R5: Write the author of this species.

A: The company was indicated.

R6: Write only month and year.

A: Corrected.

R7: It is enough to write only the common name in all subtitles, check along the text.

A: The authors agree with the Reviewer's remark. We omitted the Latin names of investigated plant extracts.

R8: Try to improve this Figure and its border

A: Dear reviewer! The authors are grateful for valuable observation. However, it turned out that the quality deteriorated during the conversion of the file from Word to PDF. We notify the Reviewer that quality will be guaranteed by providing the original version of this Figure along with the main document.

R9: Cite the following paper which reported the same method Chemical Composition and Antimicrobial Properties of Mentha _ piperita cv. ‘Kristinka’ Essential Oil. Plants  2021, 10, 1567.

A: Dear reviewer, the authors are grateful for the literature provided. The reference was included in the manuscript within lines 705 – 706. 

R10: Cite here the following paper:  Antimicrobial activity and chemical composition of essential oil extracted from Solidago canadensis L. growing wild in Slovakia. Molecules, 2019, 24, 1206; 1-13.

A: Dear reviewer! The authors are grateful for the literature provided but, unfortunately, do not see the option to rely on it since this article examines the antimicrobial activity of essential oils that have entirely different specificities of constituents (terpenes) than those discussed in the present study.

R11: Replace with "In this study, we conducted the first examination of the minimum inhibitory concentration (MIC) and inhibition zone values of industrially produced water extracts derived from various parts of the European cranberry bush (Viburnum opulus L.) (VO) including flowers, bark, berries, and a mixture thereof. These extracts were tested against 19 cultures which underwent antibiotic susceptibility tests."

A: The authors appreciate the Reviewer's suggestion. We confirm the inclusion of this sentence in the Conclusions sections, as suggested.

Reviewer 5 Report

Comments and Suggestions for Authors

Scrutinizing Antimicrobial and Antioxidant Potency of European Cranberry Bush (Viburnum opulus L.) Extracts

The authors screened the antioxidant activity and compounds of different parts of V. opulus. The work is suitable to be accepted, but the authors first must answer some aspects. Authors should also carefully review writing and language. There is no clear justification for the work and information must be added.

The authors must add consecutive line numbering to be able to be reviewed more exhaustively, it cannot be reviewed in detail.

Abstract:

Please substitute “VO” in the rest of the manuscript. It is more correct “V. opulus” than “VO”.

Add the objective of the work since there is no justification.

The justification of the work is not observed in the introduction section. Authors must provide a strong justification for the work. Additionally, an objective must be added.

In the headings and subheadings, eliminate the name of the species, it is obvious that the authors worked with their extracts. Review the entire manuscript.

In section 2.5.1. Add the concentrations used to make the calibration curve. Also mention the concentrations used of the extracts.

In sections 2.5.1 – 1.5.4. Concentrations should be in relation to the weight of the extract or biomass rather than volume. Please correct or justify. The same comment for the antioxidant activity for all the methodologies used.

Comments on the Quality of English Language

Must be checked.

Author Response

Response to Reviewer's 5 comments

R: Scrutinizing Antimicrobial and Antioxidant Potency of European Cranberry Bush (Viburnum opulus L.) Extracts

R1: The authors screened the antioxidant activity and compounds of different parts of V. opulus. The work is suitable to be accepted, but the authors first must answer some aspects. Authors should also carefully review writing and language. There is no clear justification for the work and information must be added.

A: The authors thank the Reviewer for carefully checking our manuscript and for valuable comments. The authors have incorporated most of the suggested changes in the manuscript. The authors refer to them in detail below.

R2: The authors must add consecutive line numbering to be able to be reviewed more exhaustively, it cannot be reviewed in detail.

A: Line numbering was ensured within entire manuscript. 

Abstract:

R3: Please substitute "VO" in the rest of the manuscript. It is more correct "V. opulus" than "VO".

A: The authors are grateful for the Reviewer's point. We confirm that "VO" was substituted with "V. opulus" within the entire manuscript.

R4: Add the objective of the work since there is no justification.

A: The authors are grateful for the Reviewer's remark. The objective of the work was provided: "Abstract: Considering the documented health benefits of Viburnum opulus L. (V. opulus), including anti-inflammatory and antioxidant activities, the present study was designed to qualitatively and quantitatively evaluate the biochemical profile and antimicrobial potency of four commercially available V. opulus extracts that were obtained from flowers, bark, berries and a mixture thereof by cold ultrasound-assisted extraction......"

R5: The justification of the work is not observed in the introduction section. Authors must provide a strong justification for the work. Additionally, an objective must be added.

A: The authors appreciate the Reviewer's remark. Here, we justify the work, which is also ensured in the Introduction section: "Currently, limited information is available on the effective concentrations of specific Viburnum opulus L. extracts that can inhibit or delay the growth of pathogens. Most studies have focused on antimicrobial activity testing using the Kirby-Bauer disk diffusion susceptibility method. Thus, this study's primary goal was to examine four commercially available Viburnum opulus L. extracts that were taken from (1) flowers, (2) a combination of morphological parts (flowers, berries without seeds, leaves, buds, and bark), (3) bark, and (4) berries without seeds. To achieve this, a qualitative and quantitative analysis of the bioactives profile and antimicrobial susceptibility test (MIC analysis) of reference test cultures were conducted, and the lowest concentrations that could inhibit the growth of microorganisms were determined.

R6: In the headings and subheadings, eliminate the name of the species, it is obvious that the authors worked with their extracts. Review the entire manuscript.

A: The authors agree with the Reviewer's remark. We omitted the Latin names of investigated plant extracts.

R7: In section 2.5.1. Add the concentrations used to make the calibration curve. Also mention the concentrations used of the extracts.

A: Dear reviewer! Additional information with calibration data and linearity for group compounds (2.5.1. – 2.5.4.) and antiradical activity tests (2.6.1. – 2.6.3.) is now indicated in Table S2 as Supplementary Material. The concentrations of extracts used for determining group compounds and antiradical activity are presented also in each subsection (from 2.5.1 to 2.6.3.).

R8: In sections 2.5.1 – 1.5.4. Concentrations should be in relation to the weight of the extract or biomass rather than volume. Please correct or justify. 

A: The authors appreciate the Reviewer's remark. However, the authors dare to disagree with the statement as they believe that water's mass and volume values are equal. They prefer to use the volume values because the object of the study is in a liquid state with a density equivalent to water and did not have fluctuations observed between measurements (mass w and volume V). Expressing concentrations by mass of raw materials (morphological parts of V. opulus) is impossible since there is no reliable information regarding the preparation of extracts by the manufacturer.

R9: The same comment for the antioxidant activity for all the methodologies used.

A: Dear reviewer! Please refer to our previous answer (R8).

Round 2

Reviewer 1 Report

Comments and Suggestions for Authors

No comments

Author Response

A: The authors would like to extend their heartfelt appreciation to the esteemed Reviewer who has provided invaluable suggestions, enabling us to address the manuscript’s shortcomings and further enhance its overall quality.

Reviewer 2 Report

Comments and Suggestions for Authors

I would like to thank the authors for their corrections. However, some comments are still persistent.

- Initial comment: In the results and discussion section, the authors started with the chemical composition that they used to discuss the antimicrobial composition. However, since the results of that activity were not reported yet, it was difficult to follow. To make it clearer to readers, authors can either put these ideas in the section reporting the antimicrobial activity results, or separate results and discussion into two sections.

New comment: The suggestion was not to report the antimicrobial activity results first because the chemical composition has to come first to understand the activities. However, using the chemical profile to explain the antibacterial activity while the results of that part are not given yet is illogical. The authors are discussing results that are not given yet. In this case, it's better to separate the results and discussion sections, that way the ideas used in the discussion make more sense since the results are presented first.

- The authors chose to change the order of results and discussion by putting antimicrobial activity first and then antioxidant, but I believe that the first version was better since the antioxidant results are more related to the chemical composition and TPC, TFC, TTC, and TAC. 

- Initial comment: R6: It is not the case, please revise.

- Initial comment: R8: Thank you for the reply. However, the explanation should be made to the sentence for the readers. And the abstract section has to be clear without reading the manuscript. Please add that "0.24 to 0.49 μL mL−1" refer to MIC, and explain what you mean by "than at 500.0 μL mL−1" cause it is not a MIC value.

- Lines 623-624: please merge both paragraphs.

- Initial comment : R23: please make that clear in the text.

- Initial comment : R25: Even though the data is limited since it's the first study, it is still incorrect to compare juice and extracts. I propose rephrasing the idea to show the difference without direct comparison.

- Initial comments: R26 and R39: The results are not comparable when the species are too different. At least use plants from the same genus. 

- Initial comment : R37: dear authors, you can't emphasize the superiority of investigated extracts over the other medicinal plants by citing only a few reports. The research is vast and here is only cited some plants. Please, compare what's comparable and use extracts from Viburnum genus.

Comments on the Quality of English Language

Please check the text for any possible amelioration.

Author Response

Response to Reviewer’s 2 comments

R1: I would like to thank the authors for their corrections. However, some comments are still persistent.

A: The authors would like to extend their heartfelt appreciation to the esteemed Reviewer who has provided invaluable suggestions, enabling us to address the manuscript’s shortcomings and further enhance its overall quality.

- Initial comment: In the results and discussion section, the authors started with the chemical composition that they used to discuss the antimicrobial composition. However, since the results of that activity were not reported yet, it was difficult to follow. To make it clearer to readers, authors can either put these ideas in the section reporting the antimicrobial activity results, or separate results and discussion into two sections.

R2: New comment: The suggestion was not to report the antimicrobial activity results first because the chemical composition has to come first to understand the activities. However, using the chemical profile to explain the antibacterial activity while the results of that part are not given yet is illogical. The authors are discussing results that are not given yet. In this case, it’s better to separate the results and discussion sections, that way the ideas used in the discussion make more sense since the results are presented first.

A: The authors respectfully bring to the Reviewer’s attention that they have changed the manuscript by separating the Results and Discussion sections.

R3: - The authors chose to change the order of results and discussion by putting antimicrobial activity first and then antioxidant, but I believe that the first version was better since the antioxidant results are more related to the chemical composition and TPC, TFC, TTC, and TAC. 

A: Dear Reviewer! The order of results in the manuscript has been restored to its original sequence. 

R4: - Initial comment: R6: It is not the case, please revise.

A: Corrected. Thank you.

R5: - Initial comment: R8: Thank you for the reply. However, the explanation should be made to the sentence for the readers. And the abstract section has to be clear without reading the manuscript. Please add that “0.24 to 0.49 μL mL−1” refer to MIC, and explain what you mean by “than at 500.0 μL mL−1” cause it is not a MIC value.

A: Dear Reviewer! We would like to inform the Reviewer that the authors have provided additional clarification for these values included in the Materials and Methods section’s Lines: 307-309 and Lines: 312-314. Furthermore, we have described those numbers in the abstract section.

R6: - Lines 623-624: please merge both paragraphs.

A: Dear Reviewer! Now, these paragraphs are presented as a whole in the Discussion section. Thank you.

R7: - Initial comment: R23: please make that clear in the text.

A: We have revised this sentence, and it appears as follows: “Glutamic acid is the second most prevalent amino acid (AA) identified in the V. opulus extracts. Its contribution to the total AA amount

was 5.2 to 29.2% in the V. opulus extracts. The highest glutamic acid content was

found in berry extract, while the lowest was in flower extract.” Please refer to Lines: 558-560.

R8: - Initial comment: R25: Even though the data is limited since it’s the first study, it is still incorrect to compare juice and extracts. I propose rephrasing the idea to show the difference without direct comparison.

A: Dear Reviewer! The authors revised this fragment, omitting information less related to the object of the present study. Lines: 574-584.

R9: - Initial comments: R26 and R39: The results are not comparable when the species are too different. At least use plants from the same genus. 

A: Dear Reviewer! The authors revised this fragment, omitting information less related to the object of the present study. Lines: 798-805.

R10: - Initial comment: R37: dear authors, you can’t emphasize the superiority of investigated extracts over the other medicinal plants by citing only a few reports. The research is vast and here is only cited some plants. Please, compare what’s comparable and use extracts from Viburnum genus.

A: Corrected. Thank you!

Reviewer 4 Report

Comments and Suggestions for Authors

Regarding the comment no. R10,  the mentioned reference is indicating the utilized method . as you can see in the methodology section (2.11.2. Minimum Inhibitory Concentration (MIC): 

“The MIC of each VO extract was determined using the microdilution method in 96-well plates …………………….” . So, the suggested new reference reports the same method of using Microplate-96 for determining the MIC values. Anyway, all other comments were addressed and the manuscript is improved now.

Author Response

A: The authors would like to extend their heartfelt appreciation to the esteemed reviewer who has provided invaluable suggestions, enabling us to address the manuscript's shortcomings and further enhance its overall quality.